# INTERNAGENT-MLE: NAVIGATING FINE-GRAINED OPTIMIZATION FOR CODING AGENT

## ABSTRACT

Large language models (LLMs) have shown impressive performance in general programming tasks. However, in Machine Learning Engineering (MLE) scenarios such as AutoML and Kaggle competitions, achieving high performance depends heavily on expert intervention and repeated adjustments rather than simply generating correct code. When applied directly to these tasks, LLMs often lack fine-grained domain priors, and existing MLE approaches that use linear or tree-structured searches limit knowledge transfer to adjacent hierarchical links. As a result, they cannot leverage past full trajectories or share information across branches, limiting self-evolving ability and search-space diversity. To address these limitations, we introduce InternAgent-MLE, an LLM-based coding agent that integrates a domain knowledge base for high-quality prior guidance and Monte Carlo Graph Search (MCGS) for efficient exploration. MCGS retains the tree-guided exploration of MCTS while embedding a graph structure into the expansion stage to enable dynamic path reorganization, historical trajectory reuse, and multi-solution fusion to support both self-evolution and collaborative learning. Combined with fine-grained operator sets, this design improves stability and accelerates convergence. Evaluation on the MLE-Bench shows that InternAgent-MLE achieves state-of-the-art performance in numerous dimensions, such as the average medal rate and valid submission rate, under a 12-hour budget (half the standard runtime).

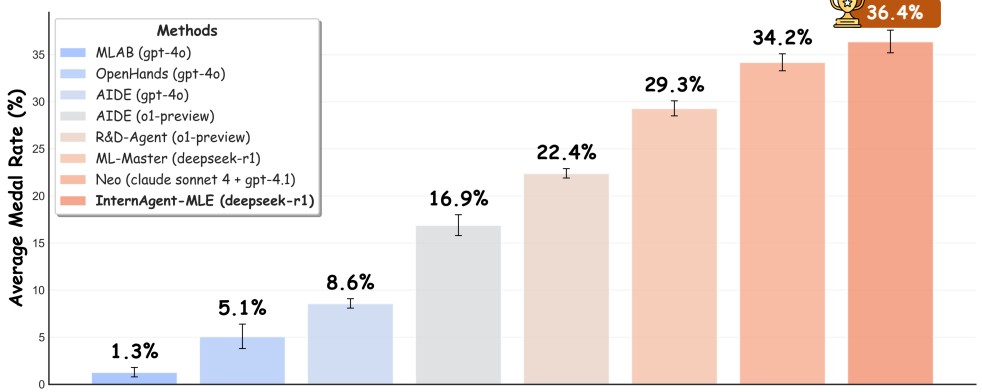

Figure 1: **The comparison across various methods on MLE-Bench.** Our InternAgent-MLE wins the championship within a 12-hour budget.

## 1    INTRODUCTION

Benefiting from the increasing capability in coding and task planning, Large Language Models (LLMs) (Hurst et al., 2024; Guo et al., 2025) are shifting from simple code assistants (Qian et al., 2024; Hong et al., 2024) to autonomous agents of sophisticated Machine Learning Engineering (MLE) (Amershi et al., 2019; Chan et al., 2025). In the realm of MLE, LLM agents are required to enhance specific metrics for the given task by iteratively optimizing code, which requires a

comprehensive consideration of various factors such as data, model architectures, and so on. While recent development of AutoML (He et al., 2021; Feurer et al., 2022) has brought about significant progress in optimizing discrete stages such as data processing, they often fall short of managing the entire end-to-end MLE workflow (*i.e.*, from data preparation to model training and inference).

Recent advancements in AI agents, have spurred the creation of MLE agents (Wang et al., 2024; Huang et al., 2023; Guo et al., 2024), which leverage the planning and execution capabilities of LLMs to optimize task performance across a broader search space. As a pioneer, AIDE (Jiang et al., 2025) reformulates the exploration process of optimizing codes as a tree search and achieves gold medals in some Kaggle competitions. R&D-Agent (Yang et al., 2025) iteratively refines codes through the cooperation of the researcher agent and the developer agent. ML-Master (Liu et al., 2025b) introduces a selectively scoped memory mechanism to integrate exploration and reasoning.

Despite the remarkable improvements on MLE tasks, existing MLE agents still suffer from the following issues. First, previous works exhibit an ***over-reliance on the internal knowledge of LLMs***. This dependence becomes a bottleneck when handling tasks in specialized domains where the internal knowledge of LLMs is often incomplete or absent. Consequently, the agent cannot integrate external domain expertise and optimize code effectively. Second, current MLE agents (Jiang et al., 2025; Liu et al., 2025b) mainly employ tree-structured search paradigms (*e.g.*, MCTS), which may lead to ***node isolation***. This issue manifests in several ways: (1) Policy updates are driven primarily by feedback from immediate parent nodes, preventing the agent from abstracting the core reasons of success or failure across an entire trajectory. (2) Search proceeds in isolated branches, inhibiting the transfer and reuse of high-quality solutions discovered in one branch by others. (3) High-quality solutions are isolated in various branches, preventing their reorganization and integration into a better solution.

Motivated by this, we propose **InternAgent-MLE**, an LLM-based coding agent that integrates a curated ML knowledge base with **Monte Carlo Graph Search (MCGS)** algorithm for MLE tasks. Specifically, the knowledge base provides domain priors across model, data, and strategy dimensions, reducing cold start errors and supporting finer-grained improvements during search. To address the isolation and limited reuse in tree search, we introduce MCGS, a variant of MCTS that incorporates graph structure into the expansion stage, allowing trajectory recall, cross-branch reference, and multi-branch aggregation. In addition, a fine-grained operator set is designed to stabilize operations and improve executability. Consequently, InternAgent-MLE achieves more stable and efficient exploration of end-to-end ML pipelines, leading to stronger solutions on challenging MLE tasks. Extensive experiments on MLE-Bench demonstrate its effectiveness, where InternAgent-MLE attains a 36.4% average medal rate under a 12-hour budget, outperforming all existing baselines.

In conclusion, our key contribution can be summarized as follows:

- We propose InternAgent-MLE framework, the first graph-search-based end-to-end MLE task solver, which couples a curated domain knowledge base with MCGS to produce complete, high-quality ML pipelines by unifying general and specialized knowledge.
- We develop Monte Carlo Graph Search (MCGS), a variant of MCTS that introduces the compositional flexibility of graphs, thereby expanding search diversity and reusability. In addition, a set of fine-grained operators are designed to stabilize execution and enhance solution quality.
- Extensive experiments on MLE-Bench show that InternAgent-MLE achieves state-of-the-art performance under a 12-hour budget, including a 36.4% average medal rate and 18.7% gold medals, outperforming all existing baselines.

## 2 RELATED WORK

### 2.1 GENERAL-PURPOSE CODING FRAMEWORKS

Recent advances in Large Language Models have led to the development of powerful LLM-based agents (Gauthier & Contributors, 2023; Wang et al., 2024; 2025) designed to tackle general software engineering tasks. Most early LLM-based agents were designed as general coding assistants, providing a flexible architecture without domain-specific tuning. For example, OpenHands (Wang et al., 2024) integrates LLM reasoning with tool use for complex software engineering tasks. SWE-Agent (Yang et al., 2024) offers comprehensive command sets for navigating codebases and implementing solutions, achieving notable performance on software engineering benchmarks. Our work

also aims to enhance the coding capabilities of LLM-based agents, but unlike these works, we focus on developing an advanced coding agent specially for ML task.

## 2.2 SPECIALIZED CODING AGENTS FOR ML ENGINEERING

To address the unique challenges of MLE, a dedicated class of coding agents has been developed (Jiang et al., 2025; Nam et al., 2025; Fang et al., 2025; Liu et al., 2025b), with many evaluated on comprehensive benchmarks like MLE-Bench (Chan et al., 2025). These agents primarily frame the problem as a search for an optimal code-based solution. Early works like AIDE (Jiang et al., 2025) employ a greedy search strategy, which can be susceptible to local optima. To overcome this, subsequent frameworks have adopted more sophisticated exploration strategies. Multi-agent collaboration approaches like AutoKaggle (Li et al., 2024) distribute tasks among specialized agents. Tree search has also emerged as a dominant paradigm. AutoMind (Ou et al., 2025) introduces an agentic tree search grounded by an expert knowledge base, while R&D-Agent (Yang et al., 2025) manages parallel exploration traces. AI auto-research agents (Team et al., 2025) systematically shows that high ML coding performance requires a careful co-design of both search policies and operators. However, these work often use isolated search paths and fail to facilitate the reuse of granular solutions. Our method resolves this inefficiency by fusing a knowledge base with MCGS to supplement task-specific knowledge and provide better recall, reference, as well as aggregation.

## 2.3 GRAPH-BASED PLANNING AND SEARCH

Early methods that combine graph structures with MCTS, often also referred to as MCGS (Czech et al., 2020; Leurent & Maillard, 2020), were developed for planning and reinforcement learning tasks with well-defined state spaces. These approaches merge identical states encountered across different branches into a single node, converting the tree into a graph to avoid redundant sampling and repeated exploration, and they require modified selection and backpropagation rules to support multi-parent nodes. In contrast, although our framework also introduces a graph-like structure, it targets an open-ended LLM-based code-generation scenario in which each node represents a distinct and diverse solution. In this setting, the purpose of MCGS is not to compress the state space but to enhance information flow and broaden exploration. Graph-based reasoning works, such as knowledge-graph multi-hop reasoning (Liu, 2025) or MCTS over compressed document graphs (Ma et al., 2024), operate on external relational structures rather than modeling the search process itself. More recently, LLM-driven coding frameworks like LocAgent (Chen et al., 2025) and CodexGraph (Liu et al., 2025a) use graphs as static dependency or codebase representations for retrieval or localization, but these graphs do not evolve during search or participate in candidate generation. In contrast, our method constructs a dynamic search graph whose reference edges enable cross-path reuse, trajectory recall, and multi-solution composition, directly addressing the limited information flow and isolated-branch behavior inherent in tree-based exploration.

## 3 INTERNAGENT-MLE

In the MLE and automated algorithm design process, strong solutions often arise from careful design, reuse of past experience, and reference of multiple candidate pathways, rather than from a single linear refinement and iteration. Tree-based search methods (Liu et al., 2025b; Toledo et al., 2025), such as MCTS, balance exploration and exploitation through branch-specific lineages, but this structure restricts knowledge flow and compositional reuse across branches and layers.

In this section, we introduce **InternAgent-MLE** as shown in Figure 2, a framework for LLM-driven ML pipeline generation. The design combines three key components: (1) a knowledge base that supplies ML domain priors and references for initialization and iterative search, (2) MCGS, which extends MCTS-based pipeline with graph edges for trajectory reuse and cross-branch integration, and (3) fine-grained operator sets that improve executability and stability.

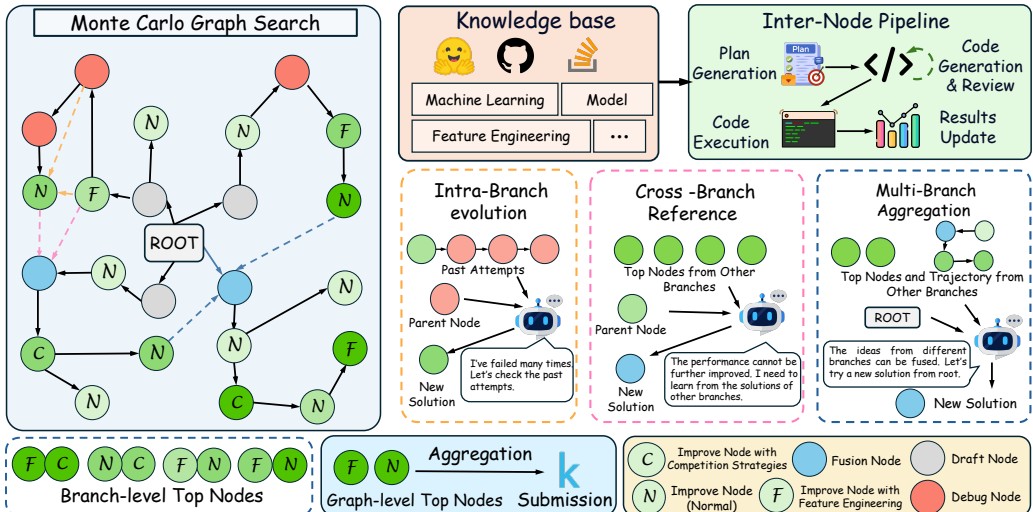

Figure 2: **The framework of InternAgent-MLE.** It consists of two main modules: (i) a curated ML domain knowledge base, and (ii) the MCGS module, which integrates graph-based exploration with a set of fine-grained operators. Detailed descriptions are provided in Section 3.

## 3.1 PROBLEM FORMULATION

Our objective is to automate the search, design, and optimization of end-to-end ML pipelines. We formalize the task as identifying the optimal solution within a search space (Jiang et al., 2025), where each node represents a complete candidate pipeline spanning preprocessing, feature engineering, model training, and prediction. The goal is to select the best-performing solution for a given task:

$$s^* = \arg\max_{s \in \mathcal{S}} h(T, s),  \tag{1}$$

where $h(T, s)$ denotes the evaluation of candidate solution $s$ on task $T$, which may vary by task (*e.g.*, accuracy, AUC, or loss). The solution space $\mathcal{S}$, often organized as a tree or graph structure, contains all possible pipelines, and the search process aims to find the solution that optimizes the task metric.

## 3.2 ML DOMAIN KNOWLEDGE BASE

Effective ML algorithm design typically relies on domain priors and hands-on experience. LLM knowledge alone is insufficient for complex tasks, leading to cold start and a high rate of early-stage errors. To address this, an ML domain knowledge base is curated and maintained, which improves the reliability of initial solutions, and provides ongoing reference during the search process.

### 3.2.1 KNOWLEDGE BASE CONSTRUCTION

We design an ML domain knowledge base across three dimensions by synthesizing practices from open-source repositories and competition platforms such as Hugging Face, GitHub, followed by careful selection. **Model-level** knowledge categorizes models by application domain and provides concise descriptions with usage guidelines, enabling the agent to quickly select and operate suitable backbones across tasks. **Data-level** knowledge summarizes modality-specific constraints and preprocessing methods, highlights key feature-engineering principles. Finally, **strategy-level** knowledge focuses on practical tricks such as test-time augmentation (TTA) and ensembling methods, distilled from discussions of ML competitions. We provide a concrete example of knowledge base structure is shown in Appendix A.1.

### 3.2.2 KNOWLEDGE RETRIEVAL AND USAGE

To preserve the agent's autonomous exploration ability, model-level knowledge is injected only during initial solution generation. Given a task $T$, the system retrieves relevant elements $R_{KB}(T)$

by matching the task description with domain keywords such as audio, natural language processing, image classification. The retrieved knowledge includes concise model descriptions and usage guidelines, serving as lightweight priors to complement LLM reasoning. It is treated as an optional signal that the agent may adopt it, use it partially, or ignore it. Formally, the initial candidate is:

$$s_{init} = \text{Init}(T, R_{KB}(T)), \tag{2}$$

where Init denotes the initialization that the agent uses to generate plan and code. During search, data- and strategy-level knowledge heuristically provides insight, enabling the agent to reason with more specific and advanced perspectives.

### 3.3 MCGS-GUIDED EXPLORATION IN MLE

In this section, we propose *Monte Carlo Graph Search (MCGS)*, which extends MCTS by incorporating a graph structure into the expansion stage via branch–node dynamic fusion. MCGS explicitly introduces trajectory recall and branch-level experience aggregation, thereby enabling more flexible composition and improved knowledge sharing.

#### 3.3.1 GRAPH-BASED SEARCH SPACE FORMULATION

To realize the optimization objective in Equation (1), we organize the search process over the solution space as a directed graph:

$$G = (V, E), \quad E = E_T \cup E_{\text{ref}}, \tag{3}$$

where the node set $V$ corresponds to candidate solutions, and each node $v \in V$ maps to a complete solution $s(v) \in \mathcal{S}$. Directed edges capture generative and reference relationships:

- **Primary edges** $E_T$: if $(u, v) \in E_T$, then node $v$ is obtained by applying an operator $o$ to node $u$ (i.e., $v = g_o(u)$). These edges preserve the parent–child generative order and are treated exactly as in classical MCTS statistics for selection and backpropagation.
- **Reference edges** $E_{\text{ref}}$: if $(r, v) \in E_{\text{ref}}$, then node $v$ obtains information from node $r$ as an extra reference beyond the parent link. Such edges connect nodes across branches or non-adjacent levels, enabling knowledge flow and compositional transfer, and they do not participate in backpropagation. When $E_{\text{ref}} = \varnothing$, the search reduces to standard tree-based MCTS.

#### 3.3.2 MCGS-BASED EXPLORATION

MCGS process follows the classical MCTS loop, retaining its strengths in selection and backpropagation, while extending the expansion phase with branch–node fusion in a dynamic graph. Through iterative exploration, the solution graph grows progressively to cover diverse candidate paths, and the best solution is returned at the stopping criterion.

**Selection.** Although overall search space is formulated as a graph, the selection stage operates solely on the tree backbone formed by primary edges $E_T$. At the beginning of each iteration, the selection policy $\pi_{\text{sel}}$ traverses $E_T$ edges in a top-down manner to identify a node $v_t$ for expansion. For a given parent node $v$, the next child is chosen from its successors $\mathcal{C}(v)$ along $E_T$ using the UCT criterion:

$$\pi_{\text{sel}}(v) = \arg \max_{i \in \mathcal{C}(v)} \text{UCT}(i), \quad \text{where } \text{UCT}(i) = \frac{Q_i}{N_i + \varepsilon} + c\sqrt{\frac{\ln(N_v + 1)}{N_i + \varepsilon}}, \tag{4}$$

where $Q_i$ denotes the accumulated reward of child node $i$, $N_i$ is its visit count, $N_v$ is the visit count of the parent node $v$, and $c > 0$ controls the strength of exploration, $\varepsilon > 0$ is a small smoothing constant to avoid division by zero. The selected node $v_t$ is then passed to expansion and evaluation.

**Expansion.** To incorporate information flow and compositional reuse into the search process, we extend the original MCTS expansion with four types of operations:

**(1) Primary expansion.** In this case, the new node is generated solely from its parent without referencing other nodes. Given the selected node $v_t$ and an operator $o \in \mathcal{O}$, expansion produces

$$v_{\text{new}} = g_o(v_t, \varnothing), \qquad (v_t, v_{\text{new}}) \in E_T, \tag{5}$$

where the reference set is empty ($R = \varnothing$), i.e., no cross-branch information is incorporated. This operation constitutes the baseline expansion, against which the graph-based variants extend. Typical operators in this form include, for example, *Draft*, *Improve*, and *Debug*, as detailed in §3.3.3.

**(2) Intra-branch evolution.** Inspired by human problem-solving strategies, this mode emphasizes reflecting on past attempts instead of blind trial and error. Practitioners review previous actions to see which changes improved outcomes or caused failures. Through self-reflection, the agent makes small adjustments, reinforcing effective patterns while avoiding repeated mistakes. Formally, given a node $v_t$, the agent takes the nearest $k$ nodes within the same branch to form a local trajectory, denoted as the intra-branch history reference set $\mathcal{R}_{\text{hist}}(v_t, k) \subseteq V$, and generates a new solution:

$$v_{\text{new}} = g_o(v_t, \mathcal{R}_{\text{hist}}(v_t, k)), \qquad (v_t, v_{\text{new}}) \in E_T, \ \ \{(r, v_{\text{new}}) \mid r \in \mathcal{R}_{\text{hist}}(v_t, k)\} \subseteq E_{\text{ref}}. \quad (6)$$

Here, $E_T$ preserves the parent–child relation, while $E_{\text{ref}}$ records the information flow from intra-branch history. The agent autonomously integrates both successful and failed experiences to form improved solutions, whereas selection and backpropagation are still conducted exclusively along $E_T$.

**(3) Cross-branch reference.** In ML competitions, contestants often draw inspiration from community-shared solutions when progress stalls. Similarly, MCGS selects a small set of high-quality nodes from other branches as references when a branch shows signs of stagnation—such as no improvement in the current branch over several consecutive expansions. Formally, at a candidate node $v_t$, a reference set $\mathcal{R}_{\text{cross}}(N)$ is formed by taking the top-$N$ nodes across all evaluated branches, ranked by performance and stability. The new candidate is then generated as

$$v_{\text{new}} = g_o(v_t, \mathcal{R}_{\text{cross}}(N)), \qquad (v_t, v_{\text{new}}) \in E_T, \ \ \{(r, v_{\text{new}}) \mid r \in \mathcal{R}_{\text{cross}}(N)\} \subseteq E_{\text{ref}}, \quad (7)$$

where $E_{\text{ref}}$ passes cross-branch knowledge, allowing agent to draw on strong solutions from other branches. Source selection and reuse intensity are determined by the agent during candidate formation.

**(4) Multi-branch aggregation.** For complex tasks, progress often requires synthesizing complementary insights from multiple strong solutions. This resembles a form of collective intelligence, where trajectories from different branches are merged and fragments of useful insights are combined to spark novel directions. When existing branches have accumulated sufficient trajectories, a new branch root is introduced beneath $v_0$, serving as a fresh starting point. This action are based on observable signals such as stagnation duration, the number of successful branches, execution feedback, and the agent's internal reasoning traces. $\mathcal{R}_{\text{agg}} = \bigcup_{b \in \mathcal{B}} \mathcal{T}_b^{\text{top}}$ denote the reference set formed by aggregating top trajectories from multiple branches, where $\mathcal{T}_b^{\text{top}}$ represent the best-performing trajectories (or nodes) in branch $b$. A new candidate is generated as

$$v_{\text{new}} = g_o(v_0, \mathcal{R}_{\text{agg}}), \qquad (v_0, v_{\text{new}}) \in E_T, \ \ \{(u, v_{\text{new}}) \mid u \in \mathcal{R}_{\text{agg}}\} \subseteq E_{\text{ref}}. \quad (8)$$

Here, $E_{\text{ref}}$ records the knowledge sources being fused. Unlike incremental refinements along a single branch, this aggregation mechanism reorganizes thoughts from diverse origins into a wholly new branch, thereby opening an independent trajectory for exploration.

**Simulation.** After generating a candidate $v_{\text{new}}$, its code is executed in an interpreter. The execution outputs are parsed to extract the task-specific metric and the execution status and written back to the node. To evaluate each node, we assign an immediate reward $R(v)$ that reflects execution validity and actual performance contribution, independent of the operator used to generate the node. Formally, the reward is defined as:

$$R(v) = \begin{cases} -1, & \text{if execution fails or no valid metric is obtained} \\ 1, & \text{if execution succeeds but does not improve the branch best} \\ 2, & \text{if execution succeeds and refreshes the branch best metric.} \end{cases} \quad (9)$$

This simple structure distinguishes failed runs, feasible but non-improving attempts, and genuine improvements, enabling stable and interpretable credit assignment during MCGS.

**Backpropagation.** After simulation, reward and status are propagated to the root only along primary edges $E_T$, while reference edges $E_{\text{ref}}$ are excluded to keep credit assignment stable and interpretable.

Each ancestor node $u$ on the branch path updates its visit count $N_u$ and cumulative reward $W_u$, which later determines its UCT value:

$$N_u \leftarrow N_u + 1, \qquad W_u \leftarrow W_u + R(v), \tag{10}$$

and the mean value is recomputed as:

$$Q_u = \frac{W_u}{N_u}. \tag{11}$$

**Memory Maintenance.** Throughout the search process, we maintain structured memory at three levels. At the node level, each node stores complete information, including its plan, code, metric, analysis, and state. At the branch level, we keep the top-$N$ nodes by metric, and at the graph level, the overall top-$N$ solutions are preserved until the end. This memory mechanism provides the basis for message passing across nodes and branches in our graph search space, while improving usability and interpretability during subsequent search and analysis.

**Parallelization.** Following R&D-Agent, we extend MCGS with asynchronous branch-parallel exploration. After expanding the root node $v_0$, multiple workers independently enter the selection stage and launch their own search traces, each proceeding with expansion and backpropagation in parallel. Candidate code executions are also run in parallel threads, further improving resource utilization and accelerating discovery of diverse high-quality solutions.

### 3.3.3 FINER-GRAINED OPERATORS

Building on AIDE, a set of finer-grained operators are defined to support graph-based exploration, and a full node–interaction diagram is provided in Appendix A.2

**Draft.** This operator generates a solution from scratch, typically at initialization under the root or when new starting points are needed. Drafting may leverage the domain knowledge base (§3.1) for warm starts and reference existing memory to reduce duplication and enhance path diversity.

**Debug.** This operator is triggered only when execution fails. It repairs faulty solutions based on error traces (*e.g.*, missing dependencies, tensor shape mismatches), applying minimal modifications until the issue is fixed or the retry limit is reached.

**Improve.** This operator family is selected once a node executes successfully and produces a valid metric to refine the solution for further improvement. It comprises three variants: **Improve-Normal**, which applies small adjustments such as switching optimizers or hyperparameter changes; **Improve-FE** (Feature Enhancement), which emphasizes data augmentation and feature engineering (*e.g.*, categorical encodings, feature aggregation); and **Improve-CS** (Competition Strategies), which introduces competition-style practices from the knowledge base (*e.g.*, pseudo-labeling, ensembling).

**Fusion.** For graph-related fusion operators (the three reference-edge types in §3.3.2), they are triggered when a branch stagnates after several rounds of failed improvement or when the global structure stabilizes. These operators merge information from multiple candidate solutions by combining primary and reference edges, leveraging historical trajectory review and branch-level experience pooling to realize self-evolution and collective intelligence.

**Code Review.** After code generation, a reviewing operator checks for data leakage, naming or import errors, and metric–task mismatches. This helps maintain node quality and prevents overfitting.

**Ensemble.** During search, a global Top-$N$ set of candidate nodes is maintained. Near termination, the best solutions are heuristically combined to produce a more robust final solution.

## 4 EXPERIMENTS

### 4.1 EXPERIMENT SETUP

**Benchmark.** All experiments are tested on MLE-Bench (Chan et al., 2025), a comprehensive benchmark introduced by OpenAI for evaluating how well AI agents perform at machine learning engineering. The full set of the MLE-Bench comprises 75 Kaggle tasks, categorized by complexity into low, medium, and high, while MLE-Bench Lite consists of a subset of 22 low-complexity tasks for teams with limited computational resources. More details are provided in Appendix A.3.

Table 1: Percentage of achieving any medals across different ML task complexity levels (left) and other evaluation dimensions (right) on MLE-Bench. Reporting results are mean ± SEM over 3 seeds; * denotes single run. Valid, Median+, and Gold indicate the percentage of submissions with valid, above median score, and gold medal; Best performances are marked in **bold**.

| Agent | Time (h) | Medal rate in different complexity | | | | Other evaluation dimensions | | |
| | | Low (%) | Medium (%) | High (%) | Avg (%) | Valid (%) | Median+ (%) | Gold (%) |
| --- | --- | --- | --- | --- | --- | --- | --- | --- |
| **MLAB** | | | | | | | | |
| gpt-4o-24-08 | 24 | 4.2±1.5 | 0.0±0.0 | 0.0±0.0 | 1.3±0.5 | 44.3±2.6 | 1.9±0.7 | 0.8±0.5 |
| **OpenHands** | | | | | | | | |
| gpt-4o-24-08 | 24 | 11.5±3.4 | 2.2±1.3 | 1.9±1.9 | 5.1±1.3 | 52.0±3.3 | 7.1±1.7 | 2.7±1.1 |
| **AIDE** | | | | | | | | |
| gpt-4o-24-08 | 24 | 19.0±1.3 | 3.2±0.5 | 5.6±1.0 | 8.6±0.5 | 54.9±1.0 | 14.4±0.7 | 5.0±0.4 |
| o1-preview | 24 | 34.3±2.4 | 8.8±1.1 | 10.0±1.9 | 16.9±1.1 | 82.8±1.1 | 29.4±1.3 | 9.4±0.8 |
| Deepseek-R1* | 24 | 27.3±0.0 | 7.9±0.0 | 13.3±0.0 | 14.7±0.0 | 78.6±0.0 | 34.6±0.0 | 8.0±0.0 |
| **R&D-Agent** | | | | | | | | |
| o1-preview | 24 | 48.2±1.1 | 8.9±1.0 | 18.7±1.3 | 22.4±0.5 | 86.1±1.1 | 32.8±1.2 | 14.4±0.5 |
| **ML-Master** | | | | | | | | |
| Deepseek-R1 | 12 | 48.5±1.5 | 20.2±2.3 | 24.4±2.2 | 29.3±0.8 | 93.3±1.3 | 44.9±1.2 | 17.3±0.8 |
| **Neo multi-agent** Claude-Sonnet 4 + GPT-4.1 | 36 | 48.5±1.5 | **29.8±2.3** | 24.4±2.2 | 34.2±0.9 | 85.8±2.2 | 40.0±0.8 | 13.8±1.8 |
| **InternAgent-MLE (ours)** Deepseek-R1 | **12** | **62.1±3.0** | 26.3±2.6 | **24.4±2.2** | **36.4±1.2** | **96.4±0.4** | **48.4±1.2** | **18.7±0.8** |

**Implementation details.** We adopt DeepSeek-R1-0528 (Guo et al., 2025) to generate plans and Python code with temperature set to 0.5. For MCGS, the simulation budget is fixed at 500 steps and the UCT exploration constant is 1.414. For the single-task test environment, we use 32 Intel(R) Xeon(R) vCPUs, 230GB of RAM, and 1 NVIDIA A800 GPU with a 12-hour time budget and averaged results over 3 random seeds. More implementation details are introduced in Appendix A.7.

**Methods for comparison.** To provide a comprehensive comparison, we evaluate InternAgent-MLE alongside both methods tested on the full set of MLE-Bench and those only tested on MLE-Bench-Lite. These include MLAB (Huang et al., 2023), OpenHands (Wang et al., 2024), AIDE (Jiang et al., 2025), R&D-Agent (Yang et al., 2025), ML-Master (Liu et al., 2025b), Neo (NEO, 2025), MLE-Star (Nam et al., 2025), MLZero (Fang et al., 2025), KompeteAI (Kulibaba et al., 2025), and AIRA-dojo (Toledo et al., 2025). We use results reported in MLE-Bench leaderboard or their paper.

### 4.2 MAIN RESULTS

**InternAgent-MLE achieves state-of-the-art performance across MLE-Bench.** As demonstrated in Table 1 and Figure 3, our proposed method, InternAgent-MLE, achieves superior performance compared to all baseline methods. Notably, InternAgent-MLE achieves an average medal rate of 36.4% and an impressive gold medal rate of 18.7%, which are the highest among all evaluated approaches. These results highlight the robustness of InternAgent-MLE across varying levels of task complexity. Specifically, InternAgent-MLE outperforms the second-best method by a significant margin in the low-complexity category (62.1% vs. 48.5%) and the score improvement in complex categories (analyzing in detail later), demonstrating its adaptability to diverse ML challenges. In addition to other evaluation dimensions, InternAgent-MLE achieves the highest valid submission rate of 96.4%, indicating its reliability in producing consistently valid results. Furthermore, InternAgent-MLE surpasses human-level performance in 48.4% of tasks, further demonstrating its ability to generalize effectively across diverse scenarios. Compared to Neo (NEO, 2025), the second-best approach, InternAgent-MLE not only demonstrates higher medal rates but also achieves these results with reduced time consumption and computational cost. For instance, while Neo requires 36 hours to achieve its performance, InternAgent-MLE achieves superior results with only 12 hours

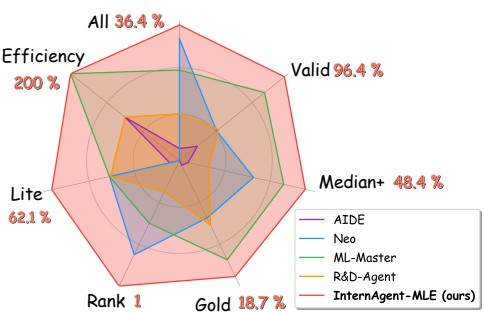

Figure 3: **MLE-Bench results of InternAgent-MLE and other methods.** It is noticeable that InternAgent-MLE performs better at all these dimensions with the shortest run time.

Figure 4: **Performance comparison on all high-level tasks of MLE-Bench.** Win means the average (3 seeds) test score of the task is better, so on for equal and lose. It can be seen that InternAgent-MLE achieves a better win rate against different baselines.

of computation time, emphasizing its efficiency and scalability. When compared with the methods tested only on MLE-Bench-Lite, InternAgent-MLE similarly achieves state-of-the-art performance (Table 2), further solidifying its position as a leading method. These results collectively highlight InternAgent-MLE's exceptional performance, efficiency, and robustness across diverse ML tasks, setting a new standard for future benchmarks and evaluations.

Table 2: Performance comparison on MLE-Bench-Lite. * means single run. Best performances are marked in **bold**.

| Agent | Medal Rate (%) |
|---|---|
| MLZero* (Claude-Sonnet 3.7) | 36.4 |
| MLE-Star (Gemini-2.0-flash) | 43.9±6.2 |
| AIRA-dojo* (o3) | 47.7 |
| KompeteAI (gemini-2.5-flash) | 51.5±1.5 |
| **InternAgent-MLE** (Deepseek-R1) | 62.1±3.0 |
| MLE-Star (Gemini-2.5-pro) | 63.6±6.0 |
| **InternAgent-MLE*** (o4-mini) | **68.2** |

**InternAgent-MLE shows a stronger ability to handle more complex problems.** In the high-level tasks of MLE-Bench, although InternAgent-MLE achieves an equivalent medal rate to the other two top-performing candidates Liu et al. (2025b); NEO (2025) in Table 1, a deeper analysis of the average task scores, illustrated in Figure 4, reveals that our method consistently outperforms the baselines across a larger number of tasks. This highlights the robustness and versatility of InternAgent-MLE when addressing the most challenging ML tasks. The higher overall scores, despite similar medal rates, indicate finer-grained optimization of our framework yields more stable and consistent improvements even in difficult scenarios.

### 4.3 ABLATION STUDY AND ANALYSIS

**Ablations on proposed components.** We conduct ablation experiments on MLE-Bench-Lite with a single seed run to evaluate the effectiveness of the proposed modules (Table 3). The *baseline* is a standard MCTS-based agent without external knowledge or graph extensions. We first add the ML domain knowledge base improves the medal rate from 40.91% to 50.00%, indicating that domain priors reduce cold-start errors and guide finer refinements. Building on this, applying intra-branch evolution of MCGS as reference edges leverages historical trajectories within the same branch, further boosting the medal rate to 59.09%. Finally, the complete framework is realized by merging cross-branch references and multi-branch aggregation, achieving a 68.12% medal rate, which demonstrates the value of reusing and reorganizing high-quality components across branches to promote both diversity and stability. In addition, Appendix A.5 reports an ablation on different knowledge-base designs, showing that model-level, data-level, and strategy-level knowledge each bring incremental gains. To further illustrate why MCG improves performance, we show some case studies demonstrating the agent's decision process under various operators in Appendix A.6. Overall, the ablation results highlight that each component contributes to the InternAgent-MLE framework's ability to handle ML tasks.

**Performance with different LLMs.** We also evaluate InternAgent-MLE across three state-of-the-art LLMs on a subset of MLE-Bench tasks: DeepSeek-R1 (Guo et al., 2025), o4-mini (OpenAI, 2025),

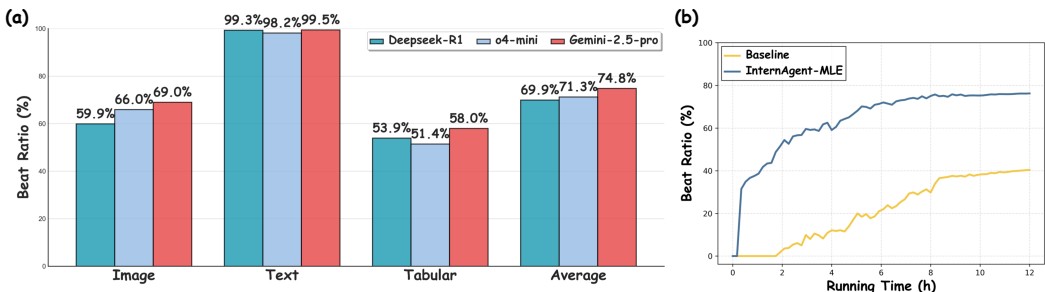

Figure 5: **(a) The comparison of different models by task type**. InternAgent-MLE is tested with different LLMs (DeepSeek-R1, o4-mini, and Gemini-2.5-pro) across image, text, and tabular tasks. **(b) The evolution of beat ratio over time**. This figure shows how InternAgent-MLE compares with the baseline under a 12-hour budget, where InternAgent-MLE consistently achieves higher leaderboard standings as search progresses.

Table 3: Ablation study on knowledge base and MCGS on MLE-Bench-Lite.

| Methods | Medal (%) | Median+ (%) | Beat (%) |
|---|---|---|---|
| baseline | 40.91% | 68.18% | 65.33% |
| + knowledge base | 50.00% | 77.27% | 68.59% |
| + knowledge base + MCGS (only Intra-branch) | 59.09% | 81.82% | 73.20% |
| **InternAgent-MLE (+ knowledge base + MCGS )** | **68.12%** | **86.36%** | **78.33%** |

and Gemini-2.5-pro (Gemini Team, 2025). As shown in Figure 5 (a), all models achieve comparable performance in text processing tasks, while showing greater variation in image and tabular domains. DeepSeek-R1 and o4-mini demonstrate similar overall performance, with Gemini-2.5-pro achieving the highest average performance. These results indicate that InternAgent-MLE scales with underlying model capacity and remains adaptable across distinct foundation models. More detailed results can be found in Appendix A.8.

**Performance over time.** To analyze the trend in the performance of InternAgent-MLE over time, we conducted an evaluation of Beat ratio vs. runtime which is presented in Figure 5 (b). As illustrated in the figure, the performance of our method improves progressively with increasing running time, which can be attributed to the proposed MCGS module's ability to interact with same/cross branch and effectively aggregate those. Furthermore, at each time step, our method consistently outperforms the baseline, demonstrating the effectiveness of the proposed components.

## 5 CONCLUSION AND DISCUSSION

In this paper, we present InternAgent-MLE, an LLM-based agent that combines a curated ML knowledge base with MCGS to address key limitations of current MLE approaches. The knowledge base provides domain priors across model, data, and strategy dimensions, improving cold-start performance and guiding finer-grained refinements. MCGS transforms the tree-structured search space into a graph, introducing trajectory recall and branch-level aggregation to support self-evolving and collective intelligence. Together with a set of specialized operators, these components enable more stable, efficient, and diverse exploration of end-to-end ML pipelines. Evaluation on MLE-Bench shows that InternAgent-MLE achieves 36.4% average medal rate under only a 12-hour budget, outperforming all existing baselines. Additional experiments further confirm the effectiveness of MCGS and the curated knowledge base across diverse tasks. While MCGS brings consistent gains in most settings, we also observe that its improvements may occasionally be limited when the underlying LLM is unable to produce any high-quality feasible candidates across branches in some hard tasks. In the future, we will incorporate multi-step, decomposed code generation to handle more complex AI tasks, and extend InternAgent-MLE to broader benchmarks beyond MLE-Bench.

ETHICS STATEMENT

This manuscript does not touch upon any topics or experiments related to ethics.

REPRODUCIBILITY STATEMENT

This manuscript provides detailed implementation details and hyperparameter settings for reproduction. In addition, the code will be open-sourced for acceptance in the future.

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

## A    APPENDIX

### USE OF LLMS

We use large language models (LLMs) only to assist in drafting and refining our manuscripts, helping improve clarity and coherence.

### A.1    EXAMPLE OF KNOWLEDGE BASE

```
Model-level Knowledge Base

"NLP": {
    "DeBERTa-v3-large": {
        "Code_template": "
            # Load tokenizer and model
            tokenizer = AutoTokenizer.from_pretrained("microsoft/deberta-v3-large")
            model = AutoModelForSequenceClassification.from_pretrained("microsoft/deberta-v3-large", num_labels=2)

            # Prepare input text
            inputs = tokenizer("Hello, my dog is cute", return_tensors="pt")

            # Inference without gradient computation
            with torch.no_grad():
                logits = model(**inputs).logits

            # Get predicted class
            predicted_class_id = logits.argmax().item()
            print(model.config.id2label[predicted_class_id])

            # To train on `num_labels` classes, set num_labels when loading the model
            num_labels = len(model.config.id2label)
            model = AutoModelForSequenceClassification.from_pretrained("microsoft/deberta-v3-large", num_labels=num_labels)

            # Prepare labels for training
            labels = torch.tensor([1])

            # Calculate training loss
            loss = model(**inputs, labels=labels).loss
            print(round(loss.item(), 2)) ",

        "Description": "
        DeBERTa-v3-large is a large-scale pretrained language model developed by Microsoft for natural language processing tasks. The model is
        typically fine-tuned by adding a linear classification or regression head on top of the [CLS] embedding or pooled features.

            Domain: natural language text. It is designed to handle a wide range of language understanding applications, including text classification,
            sentiment analysis, natural language inference, question answering, and sequence labeling. The architecture is based on the Transformer
            encoder with 24 layers, hidden size of 1024, 40 attention heads, and approximately 435 million parameters. Compared to earlier BERT and
            RoBERTa models, DeBERTa introduces disentangled attention mechanisms, improved mask decoders, and enhanced training efficiency.

            Input: tokenized text sequences of shape (batch_size, seq_len).

            Output: contextual embeddings of shape (batch_size, seq_len, hidden_dim) or classification logits when a task-specific head is attached."
    }
}
```

Figure 6: A model-level example of knowledge base.

### A.2    DETAILED OPERATIONAL LOGIC OF MCGS

Figure 7 shows the detailed operational logic and illustrative diagrams of node interactions.

### A.3    MLE-BENCH BENCHMARK

Machine Learning Engineering (MLE) represents a critical frontier in AI development, requiring sophisticated integration of coding, experimentation, and problem-solving skills. Researchers usually evaluate such capacity of an LLM agent on MLE-bench proposed by OpenAI.

Our work is also carried out on this benchmark. We now introduce MLE-Bench in detail:

MLE-bench is a comprehensive benchmark designed to assess autonomous ML engineering performance through real-world competitions. It comprises 75 carefully curated Kaggle competitions spanning diverse domains, including natural language processing, computer vision, signal processing, and tabular data analysis. These competitions are selected from 586 candidates through rigorous manual screening by ML engineers, ensuring each task represents authentic, challenging ML engineering work relevant to contemporary practice. The dataset includes competitions of varying complexity: 22

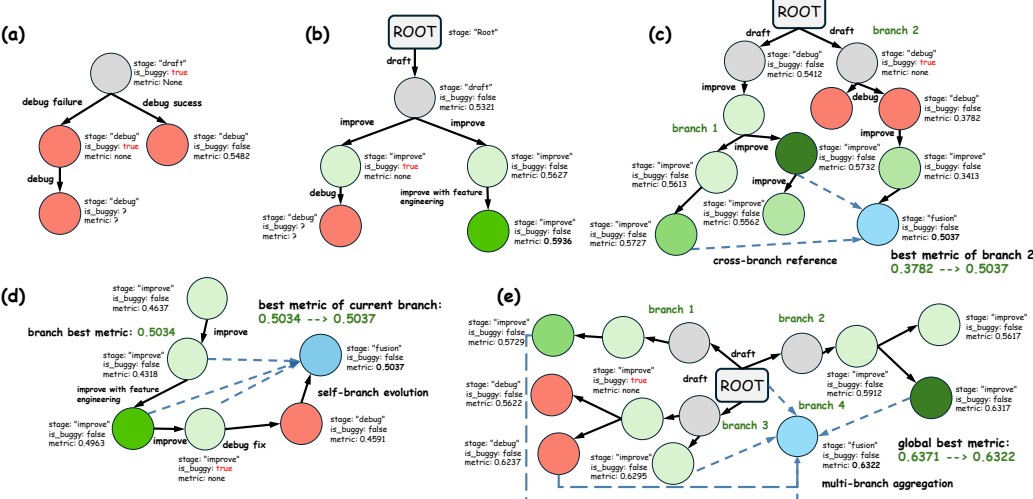

Figure 7: Node interaction and information-flow diagram of the MCGS search process. (a) – (e) show examples of debug, improve, cross-branch reference, self-branch evolution, and multi-branch aggregation under MCGS.

low-complexity tasks (solvable by experienced engineers in under 2 hours), 38 medium-complexity tasks (2-10 hours), and 15 high-complexity tasks (over 10 hours), covering 15 distinct problem categories. Each competition in MLE-bench includes the original problem description, datasets with reconstructed train-test splits, local grading code, and human baseline performance from Kaggle leaderboards. This setup enables direct comparison between AI agents and human competitors while maintaining evaluation integrity. The benchmark employs medal achievement rates as the primary metric, where agents must reach bronze, silver, or gold medal thresholds based on their performance relative to human participants. The benchmark evaluates end-to-end ML engineering capabilities, including data preprocessing, model architecture design, hyperparameter tuning, training optimization, and debugging. Agents must work autonomously within time constraints (24-hour time limit) to produce valid submission files. This comprehensive evaluation framework reveals both the promise and limitations of current AI systems in performing complex ML engineering tasks, providing crucial insights for the development of more capable autonomous ML systems.

## A.4 METRIC FOR EVALUATION

In this section, we introduce the key metrics used to assess the performance of our agent. These metrics are similar to those used by humans in Kaggle competitions. Each metric we used in the main paper is summarized below:

- **Average Medal Rate (Avg, in %):** represents the average number of task submissions that can win the medal, including silver, bronze, and gold. The threshold for the score that can earn a medal is officially provided by Kaggle and MLE-Bench.

- **Valid Submission Rate (Valid, in %):** represents validity rate of the submitted results. The submission format and other validity checks are officially provided by Kaggle and MLE-Bench.

- **Above Median Rate (Median+, in %):** represents the average number of task submissions that can beat half of the human competitors. The threshold for the score that can beat half of the human competitors is officially provided by Kaggle and MLE-Bench.

- **Gold Medal Rate (Gold, in %):** represents the average number of task submissions that can win the gold medal. The threshold for the score that can earn the gold medal is officially provided by Kaggle and MLE-Bench.

- **Agent Runtime (Time in % or Efficiency):** represents the work time for agents to produce submission files. Less running time means higher efficiency.

Table 4: Ablation study on knowledge base design.

| Knowledge Base Design | learning-agency-lab-automated-essay-scoring-2 ($\uparrow$) | stanford-covid-vaccine ($\downarrow$) |
|---|---|---|
| w/o knowledge | 0.8109 | 0.27584 |
| + model-level knowledge | 0.82921 | 0.23790 |
| + model & data-level knowledge | 0.83304 | 0.23225 |
| **InternAgent-MLE (full KB)** | **0.84965** | **0.22527** |

- **Above Beat Ratio (Average Beat, in %):** represents the average percentage of human competitors whose performance is surpassed by the task submission results. The top percentage of each score for the contestants (i.e., the beat ratio) is officially provided by Kaggle and MLE-Bench.

## A.5 ADDITIONAL ABLATION ON KNOWLEDGE-BASE DESIGN

To further analyze the contribution of different knowledge components, we conduct an ablation study on the internal structure of the knowledge base (KB). As shown in Table 4, Introducing only model-level knowledge already provides noticeable gains by supplying architecture-specific priors such as input/output formats and common training configurations. Adding data-level knowledge brings additional improvements by guiding preprocessing and feature-handling choices. The full KB, which further incorporates strategy-level engineering priors, achieves the highest performance across both tasks. These results confirm that the KB is modular, each layer contributes positively, and the full design offers the most stable and general enhancement.

## A.6 CASE STUDY

As shown in Figure 8, 9, 10, 11, 12, 13, and 14, we provide several representative cases that demonstrate InternAgent-MLE's autonomous ML engineering capabilities across different domains. These cases showcase the complete workflow from problem understanding to solution implementation, highlighting the agent's reasoning process, code generation, and performance optimization strategies.

## A.7 HYPERPARAMETERS

We provide the default hyperparameter configuration used in our MCGS framework (Table 5). These hyperparameters are used throughout all experiments unless otherwise specified, and can be tuned to adapt the algorithm to different domains or computational budgets.

## A.8 MORE DETAILED RESULTS OF DIFFERENT LLMS

To provide a comprehensive evaluation of InternAgent-MLE's adaptability across different foundation models, we conducted experiments using three state-of-the-art LLMs: DeepSeek-R1 (Guo et al., 2025), o4-mini (OpenAI, 2025), and Gemini-2.5-pro (Gemini Team, 2025). We selected a representative subset of 10 tasks from MLE-Bench, covering three distinct domains: image processing, text analysis, and tabular data tasks. Table 6 presents the detailed performance comparison across all three models on these selected tasks. The results reveal several interesting patterns:

**Image Tasks:** The performance varies significantly across models in image-related tasks. DeepSeek-R1 achieves the best performance on dog breed identification (Logloss: 0.3003) and histopathologic cancer detection (AUC: 0.9981), while Gemini-2.5-pro excels in catheter line classification (AUC: 0.9403) and document denoising (RMSE: 0.0165). This variation suggests that different LLMs may generate distinct approaches or architectures for computer vision problems, leading to varying effectiveness in the resulting ML solutions.

**Text Tasks:** All three models demonstrate remarkably consistent performance in text processing tasks, with minimal differences in scores. Gemini-2.5-pro slightly outperforms others across all text tasks, achieving the best results in toxic comment classification (AUC: 0.9879), author identification (Logloss: 0.2113), and insult detection (Accuracy: 0.9470). The small performance gaps indicate

### Generation Case

"Task": "the-icml-2013-whale-challenge-right-whale-redux"

"Thinking and Plan": "We'll use the MuQ-large-msd-iter **pre-trained model** for **audio feature extraction**, leveraging its robust music-domain representations which may transfer well to whale call detection. Audio clips will be resampled to 24kHz and processed through MuQ to obtain clip embeddings via mean pooling. A simple **logistic regression classifier** will be trained on these embeddings. For variable-length test clips, we'll split them into 2-second windows (matching training clip duration), extract features per window, and use max pooling of predictions to determine the final clip probability. This approach efficiently handles audio characteristics while maintaining computational feasibility."

```python
...

# Load MuQ model
muq = MuQ.from_pretrained("OpenMuQ/MuQ-large-msd-iter").to(DEVICE).eval()
def extract_features(audio_path, is_test=False):
    """Extract MuQ features from audio clip"""
    try:
        # Load and resample audio
        y, _ = librosa.load(audio_path, sr=SAMPLE_RATE, mono=True)

        # For test clips: split into 2-second windows
        if is_test:
            window_size = int(SAMPLE_RATE * DURATION)
            windows = [y[i:i+window_size] for i in range(0, len(y), window_size)]
            # Pad last window if needed
            windows = [np.pad(w, (0, max(0, window_size - len(w)))) if len(w) < window_size else w for w in windows]
        else:
            windows = [y]
        features = []
        for wav in windows:
            wav_tensor = torch.tensor(wav, dtype=torch.float32).unsqueeze(0).to(DEVICE)
            with torch.no_grad(), autocast():
                output = muq(wav_tensor)
            # Mean pooling of last hidden state
            pooled = output.last_hidden_state.mean(dim=1).cpu().numpy()
            features.append(pooled)

        return np.vstack(features) if is_test else pooled
# Prepare training data
...
# Train classifier with validation split
clf = LogisticRegression(max_iter=1000)
clf.fit(X_train, y_train)
...
```

Figure 8: Case of code generation.

```
### Debugging Case
"Task": "tgs-salt-identification-challenge"

"Error Info": "

model = UNet().to(device)
criterion = nn.BCELoss()
...
best_mAP = 0.0
for epoch in range(EPOCHS):
    model.train()
    train_loss = 0.0
    for images, masks in train_loader:
        images, masks = images.to(DEVICE), masks.to(DEVICE)
        optimizer.zero_grad()
        with autocast():
            outputs = model(images)
            loss = criterion(outputs, masks)
            ...
RuntimeError: torch.nn.functional.binary_cross_entropy and torch.nn.BCELoss are unsafe to autocast.

"Thinking and Plan": "The previous implementation encountered an issue where BCELoss was used with autocast,
which is unsafe. The fix involves switching to BCEWithLogitsLoss for numerical stability and removing the sigmoid
activation from the model's final layer. During inference, sigmoid is applied to convert logits to probabilities.
Additionally, the validation mAP calculation now correctly handles logits by applying sigmoid before processing."

"Fixed Code":

model = UNet().to(DEVICE)
criterion = nn.BCEWithLogitsLoss()  # Fixed loss function
...
best_mAP = 0.0
for epoch in range(EPOCHS):
    model.train()
    train_loss = 0.0
    for images, masks in train_loader:
        images, masks = images.to(DEVICE), masks.to(DEVICE)
        optimizer.zero_grad()
        with autocast():
            outputs = model(images)
            loss = criterion(outputs, masks)  # Now safe with autocast
```

Figure 9: Case of debugging process.

```
### Debugging Case
"Task": "dogs-vs-cats-redux-kernels-edition"

"Error Info": "

for i, record_id in enumerate(record_ids):
    orig_mask = torch.tensor(preds[i]).unsqueeze(0).unsqueeze(0).float()
    orig_mask = (
        torch.nn.functional.interpolate(
            orig_mask, size=(256, 256), mode="nearest"
        )
        .squeeze()
        .numpy()
        .astype(np.uint8)
        )
    rle = rle_encode(orig_mask)
    results.append({"record_id": record_id, "encoded_pixels": rle})

ValueError: The truth value of an array with more than one element is ambiguous.

"Thinking and Plan": "The previous implementation had two main issues: the RLE encoding function failed when the
runs array was non-empty but had multiple elements, and the model used an insufficient number of input bands (only
3 bands at a single time step). To fix this:
1. The RLE encoding function now correctly checks for empty arrays using `runs.size == 0`
2. The model now uses all 9 spectral bands (8-16) at the center time step
3. The first convolutional layer was modified to accept 9 input channels
4. Increased training epochs to 5 for better performance
5. Maintained other optimizations like mixed precision training"

"Fixed Code":

# Fixed RLE Encoding
def rle_encode(mask):
    pixels = mask.flatten()
    pixels = np.concatenate([[0], pixels, [0]])
    runs = np.where(pixels[1:] != pixels[:-1])[0] + 1
    runs[1::2] -= runs[::2]
    if runs.size == 0:
        return '-'
    else:
        return ' '.join(str(x) for x in runs)
```

Figure 10: Case of debugging process.

### Improving Case
"Task": "dogs-vs-cats-redux-kernels-edition"

"Thinking and Plan": "The current solution uses EfficientNetV2-M and has achieved a validation log loss of 0.057. However, I think we can improve by using a more advanced model architecture. Specifically, I propose to use the newer ConvNeXt model, which has shown state-of-the-art performance in image classification. Additionally, we can incorporate test-time augmentation (TTA) to reduce overfitting and improve generalization.

Improvement Sketch:
1. Replace the EfficientNetV2-M model with a ConvNeXt Large model, which has demonstrated superior performance on ImageNet and other benchmarks.
2. Implement test-time augmentation (TTA) during the testing phase to average predictions over multiple augmented versions of each test image, thereby improving robustness.

This change is expected to leverage a more powerful architecture and TTA to capture better features and reduce variance, leading to improved log loss.

Now, I will implement these changes in the code. Note: We must be cautious about the runtime and memory constraints."

"Improved Code":

```python
# Test predictions with TTA
test_preds = [0.0] * len(test_files)
test_ids = [os.path.basename(f).split(".")[0] for f in test_files]
tta_loaders = []
for transform in test_transforms[:TTA_STEPS]:
    tta_dataset = DogCatDataset(test_files, transform=transform, mode="test")
    tta_loaders.append(DataLoader(
        tta_dataset, batch_size=BATCH_SIZE, shuffle=False, num_workers=4
    ))
# Aggregate predictions over TTA steps
with torch.no_grad():
    for loader in tta_loaders:
        for images, img_ids in loader:
            images = images.to(device)
            outputs = model(images)
            # Average TTA predictions
            ...
test_preds = [p / TTA_STEPS for p in test_preds]
```

Figure 11: Case of improving process.

### Intra-branch evolution Case
"Task": "aptos2019-blindness-detection"

"Thinking and Plan": "Looking at the **historical trajectory**:
Step 1: Successfully implemented weighted loss in XGBoost to address class imbalance, achieving a validation kappa of 0.7332.
Step 2: Attempted to incorporate DINOv2 backbone and focal loss but encountered an AttributeError due to a naming conflict.
Step 3: Fixed the naming conflict but then faced a ValueError during validation because the dataset wasn't returning labels correctly.
Step 4: Fixed the dataset issues and achieved a validation kappa of 0.4967 using DINOv2 with a simple classifier head.
Step 5: Tried to enhance the classifier head with a deeper MLP and add early stopping/learning rate scheduling, but encountered a TypeError due to an invalid verbose argument in ReduceLROnPlateau.
Step 6: Fixed the verbose argument issue and achieved a validation kappa of 0.6561 with the enhanced classifier.
The current solution uses DINOv2 with a deeper MLP classifier, focal loss with class weights and label smoothing, and test-time augmentation. The best validation kappa achieved is 0.6561.

To improve upon the current solution, I propose to **incorporate a more advanced feature extraction technique** by using the DINOv2 model as a frozen feature extractor and fine-tuning only the classifier head. This approach can potentially lead to better generalization by leveraging the powerful pre-trained features without overfitting the entire model to the limited medical dataset. Additionally, I will adjust the training strategy by using a higher initial learning rate for the classifier and a longer training schedule with early stopping to allow the model to converge more effectively.

Now I will implement this improvement in code. The implementation will involve:
1. Freezing the DINOv2 backbone and only training the classifier head.
2. Adjusting the learning rate for the classifier to be higher (1e-3) since it is the only part being trained.
3. Increasing the number of epochs to 100 to allow for more training time.
4. Keeping the early stopping mechanism with patience of 5 epochs to prevent overfitting.
5. Using the same focal loss and TTA setup as before."

"Improved Code":

```python
# Enhanced DINOv2 model with frozen backbone
class DinoClassifier(nn.Module):
    def __init__(self, num_classes=5):
        super().__init__()
        self.dinov2 = Dinov2Model.from_pretrained("facebook/dinov2-base")

        # Freeze backbone parameters
        for param in self.dinov2.parameters():
            param.requires_grad = False
        self.classifier = nn.Sequential(
            nn.Linear(768, 512),
            nn.BatchNorm1d(512),
            nn.ReLU(),
            nn.Dropout(0.5),
            nn.Linear(512, 256),
            nn.ReLU(),
            nn.Dropout(0.3),
            nn.Linear(256, num_classes),
        )
```

Figure 12: Case of intra-branch evolution process.

### Cross-branch reference Case

"Task": "mlsp-2013-birds"

"Thinking and Plan": "must be thoughtful and selective, choosing only techniques that genuinely enhance performance rather than simply combining approaches.
After reading the reference solutions, I notice that they have achieved significantly higher AUC scores (0.8670, 0.8655, 0.85772, 0.83143) compared to my current solution (0.7387). Therefore, I need to **incorporate some of the advanced techniques** used in these solutions.

Key techniques from the reference solutions:
1. Using EfficientNet-B3 (Branch ID 2) for spectrogram feature extraction instead of MobileNetV2. EfficientNet is known for better performance in image tasks.
2. Processing full MFCC sequences using 1D convolutions and GRU layers to capture temporal patterns (Branch ID 2). This is a significant improvement over using static features (aggregated segment and histogram features) because audio data has a temporal dimension.
3. Implementing attention-based fusion of image and audio features (Branch ID 2) to better combine different modalities.
4. Per-species binary classifiers to handle class imbalance (Branch ID 5 and 2). This is important for multi-label problems with varying class frequencies.
5. Label smoothing to handle noisy annotations (Branch ID 2).
6. Dynamic learning rate scheduling (Branch ID 2 and 5).

Given the time constraints (only 16 minutes remaining), I cannot implement all of these. However, I can focus on the most impactful changes:
- **Replace MobileNetV2 with EfficientNet-B3** for spectrogram feature extraction (if time permits, but note that EfficientNet is larger and might be slower).
- Instead of aggregated features, use MFCC sequences and process them with a temporal model (1D CNN + GRU). This might require a different approach to feature extraction and model architecture.
- However, note that the current solution uses aggregated features (segment, histogram, and CNN features) and the reference solutions that achieved high AUC used temporal modeling. Therefore, **I should shift to temporal modeling.**
Considering the time constraint (16 minutes), I must design a solution that runs quickly. Extracting MFCCs and training a temporal model might take too long. Therefore, I need to balance between model complexity and runtime. Alternatively, I can use precomputed MFCCs if available? But the dataset doesn't provide MFCCs. We have to compute them from the WAV files.
Given the time, I might not be able to compute MFCCs for all audio files and train a temporal model in 16 minutes. Therefore, I'll focus on **improving the current feature-based model** by incorporating the following from the references ..."

"Improved Code":

```
# Load CNN features from spectrograms using EfficientNet-B3
def load_cnn_features(rec_id2filename):
cnn_features = {}
model = models.efficientnet_b3(pretrained=True)
model.classifier = nn.Sequential() # Remove classification head
model = model.to(device).eval()

spectrogram_dir = "input/supplemental_data/filtered_spectrograms"
dataset = SpectrogramDataset(rec_id2filename, spectrogram_dir, transform=transform)
loader = DataLoader(dataset, batch_size=32, shuffle=False, num_workers=4)
...
```

Figure 13: Case of cross-branch reference process.

### Multi-branch aggregation Case

"Task": "Stanford-covid-vaccine"

"Thinking and Plan": "The **key insights from each branch** are:
- Branch 1: Used multi-scale dilated convolutions to capture long-range dependencies and test-time augmentation with reverse complement sequences. Also incorporated experimental error weighting in the loss function and enhanced residual connections. Achieved a validation MCRMSE of 0.5776.
- Branch 3: Used a Transformer-based architecture to model long-range dependencies. Fixed a dimension mismatch issue in the linear layer. Achieved a validation MCRMSE of 0.7559.
- Branch 5: Implemented a transformer-based architecture to replace LSTM layers, adding multi-head self-attention for global sequence relationships. Increased model capacity with more layers and attention heads, and implemented residual connections and layer normalization. Achieved a validation MCRMSE of 0.7623.
- Branch 2: Added Feature-wise Linear Modulation (FiLM) layers to dynamically modulate feature representations using conditioning information (structure and loop types). This allows for context-dependent feature representations. Achieved a validation MCRMSE of 0.8973.

To create a novel solution, I will combine the strengths of these approaches:

1. **Transformer Backbone**: From Branch 3 and 5, I will use a transformer architecture as the core to capture long-range dependencies in RNA sequences. This is crucial because RNA structures involve interactions between distant nucleotides.
2. **FiLM for Feature Modulation**: From Branch 2, I will **incorporate FiLM layers** to condition the model on structural context (structure and loop types). This will help the model adapt its predictions based on the structural environment of each nucleotide.
3. **Multi-Scale Processing**: Inspired by Branch 1's dilated convolutions, I will **add convolutional layers** at multiple scales to capture local patterns that complement the global attention of transformers.
4. **Error-Weighted Loss**: Like Branch 1, I will weight the loss function by experimental errors to focus learning on more reliable data points.
5. **Test-Time Augmentation**: Also from Branch 1, I will implement reverse complement augmentation at test time to improve prediction robustness.
6. **Enhanced Residual Connections**: I will incorporate residual connections throughout the model to facilitate training deep architectures.

Now, I'll write the code accordingly, ensuring it's **self-contained and includes all necessary components**. The code will:
1. Import required libraries
2. Load and preprocess the data
3. Define the model architecture
4. Set up training and validation loops
5. Train the model
6. Make predictions on the test set with TTA
7. Save predictions to `submission/submission.csv`
8. Print the validation MCRMSE  ..."

"Improved Code":

```
# Custom loss with experimental error weighting
def weighted_mse_loss(pred, target, errors):
    mask = ~torch.isnan(target)
    loss = (pred[mask] - target[mask]) ** 2
    weights = 1 / (errors[mask] + 1e-2)
    return (loss * weights).mean()
...
```

Figure 14: Case of Multi-branch aggregation process.

Table 5: MCGS Hyperparameter Configuration.

| Hyperparameter | Description | Default |
|---|---|---|
| *General Search* | | |
| max_steps | Max search steps | 500 |
| exploration_constant | UCT exploration constant $C$ | 1.414 |
| temperature | LLM decoding temperature | 0.5 |
| max_parallel_workers | Max parallel workers | 3 |
| max_draft_num | Max Draft attempts from root | 7 |
| max_debug_num | Max Debug attempts | 20 |
| *Memory* | | |
| branch_top_N | Top-$N$ candidates kept per branch | 5 |
| global_top_N | Top-$N$ solutions kept globally | 10 |
| *Reference / Fusion* | | |
| max_history_num | Max historical trajectories used in intra-branch | 7 |
| max_ref_num | Max reference solutions used in cross-branch | 7 |
| max_agg_num | Max aggregation trajectories used in multi-branch | 7 |
| ensemble_num | Final ensemble size | 6 |
| *Knowledge base* | | |
| kb_init_ref_prob | Heuristic probability of KB reference at initialization | 0.8 |

Table 6: Score comparison on 10 MLE-Bench tasks. Best result for each task is highlighted in **bold**.

| Task | Metric | DeepSeek-R1 | o4-mini | Gemini-2.5-pro |
|---|---|---|---|---|
| *Image Tasks* | | | | |
| dog-breed-identification | Logloss ↓ | **0.3003** | 0.3941 | 0.3418 |
| ranzcr-clip-catheter-line-classification | AUC ↑ | 0.9162 | 0.9040 | **0.9403** |
| histopathologic-cancer-detection | AUC ↑ | **0.9981** | 0.9940 | 0.9980 |
| denoising-dirty-documents | RMSE ↓ | 0.0418 | 0.0181 | **0.0165** |
| *Text Tasks* | | | | |
| jigsaw-toxic-comment-classification | AUC ↑ | 0.9873 | 0.9869 | **0.9879** |
| spooky-author-identification | Logloss ↓ | 0.2163 | 0.2534 | **0.2113** |
| detecting-insults-in-social-commentary | Accuracy ↑ | 0.9391 | 0.9388 | **0.9470** |
| *Tabular Tasks* | | | | |
| new-york-city-taxi-fare-prediction | RMSE ↓ | 5.7589 | 6.2157 | **4.6956** |
| nomad2018-predict-transparent-conductors | RMSLE ↓ | **0.0585** | 0.0591 | 0.0593 |
| tabular-playground-series-may-2022 | Accuracy ↑ | **0.9796** | 0.9690 | 0.9793 |

that all three LLMs possess strong capabilities in generating effective NLP solutions, likely due to their inherent understanding of text processing methodologies.

**Tabular Tasks:** Similar to image tasks, tabular data processing shows notable performance variations. Gemini-2.5-pro demonstrates superior performance in taxi fare prediction (RMSE: 4.6956), while DeepSeek-R1 achieves the best results in material property prediction (RMSLE: 0.0585) and the playground series classification task (Accuracy: 0.9796). These differences may reflect varying approaches to feature engineering, model selection, or hyperparameter optimization generated by different LLMs.

These results confirm that InternAgent-MLE is successfully adapting to different LLMs as backends for generation. The consistent performance across text tasks and the model-specific advantages in the image and tabular domains demonstrate that different LLMs bring their unique problem solving approaches to the generation of automated machine learning solutions, while InternAgent-MLE effectively harnesses these diverse capabilities.

Table 7: Effect of `branch_top_N` on task performance.

| branch_top_N | random-acts-of-pizza (↑) | stanford-covid-vaccine (↓) |
|---|---|---|
| 1 | 0.6381 | 0.3056 |
| 5 | **0.7298** | **0.2253** |
| 10 | 0.7286 | 0.2465 |

Table 8: Effect of `max_ref_num` on task performance.

| max_ref_num | random-acts-of-pizza (↑) | stanford-covid-vaccine (↓) |
|---|---|---|
| 1 | 0.5725 | 0.3490 |
| 5 | 0.7222 | 0.2465 |
| 7 | **0.7298** | **0.2253** |
| 9 | 0.7109 | 0.2472 |

### A.9 HYPERPARAMETER SENSITIVITY ANALYSIS

To assess whether InternAgent-MLE 's performance depends heavily on specific hyperparameter choices, we conduct sensitivity experiments on two representative parameters: `branch_top_N` and `max_ref_num`. As shown in Table 7 and Table 8, the overall performance remains stable across a wide range of values. We observe degradation only when values become excessively large—introducing noisy or irrelevant candidates—or excessively small, which restricts effective cross-branch information flow. These results confirm that the main performance gains stem from the search structure rather than sensitivity to individual hyperparameters.

### A.10 COMPUTATIONAL COST ANALYSIS

Although MCGS introduces richer contextual signals during expansion, which slightly increases input tokens per LLM call, it does not increase overall computational overhead. Pure MCTS often generates low-quality candidates, repeatedly encountering failing nodes and invoking the LLM multiple times for debugging without reaching execution or training. In contrast, MCGS improves solution success rate via cross-branch information flow, allowing more candidates to reach the execution stage—the most time-consuming part of MLE tasks. As a result, MCGS reduces total LLM usage under the same 12-hour budget while achieving more effective search coverage. Table 9 compares average token usage and solution success rate across representative tasks. MCGS increases success rate from 57.3% to 66.4% while reducing both input and output tokens, demonstrating better efficiency.

Table 9: Comparison of token usage and solution success rate between MCTS and MCGS.

| Method | # Input tokens | # Output tokens | Solution success rate |
|---|---|---|---|
| InternAgent-MLE (MCTS) | 854K | 387K | 57.3% |
| InternAgent-MLE (MCGS) | 720K | 282K | 66.4% |

