# OpenReview forum: "InternAgent-MLE: Navigating Fine-Grained Optimization for Coding Agent"
_ICLR.cc/2026/Conference — Submitted to ICLR 2026_

### Official Review · Reviewer_WoZe · 2025-10-24

**Soundness:** 3
**Presentation:** 4
**Contribution:** 2
**Rating:** 4
**Confidence:** 3

**Summary:**

This paper introduces InternAgent-MLE, a coding agent that helps automate machine learning engineering tasks such as those found in Kaggle competitions. The main idea is to improve how LLM agents explore and use past work: Instead of a normal tree search (which treats each attempt in isolation), the authors propose a graph-based search method (MCGS) that lets the agent recall previous attempts and combine ideas from different solution paths, also learning from failures. They also add a curated knowledge base containing common ML tricks and model guidelines. Together, this mix of elements achieves respectable performance on the MLE benchmark. The results look solid, but the approach depends on manual curation of the knowledge base and only tests on a single benchmark. Similar recent papers (e.g., MLE-STAR) report comparable or better results, so the novelty may be smaller than claimed. Overall, this is a technically strong and carefully executed system paper, but its contribution is more incremental than groundbreaking (which it would have been, one ICLR ago).

**Strengths:**

Paper deals with a clear and relevant limitation of current MLE agents (poor reuse and isolation in tree-based searches).
It has a somewhat novel combination of graph-based search (MCGS) and domain knowledge base, with SOTA (at the time it was written, unfortunately recently scooped by other approaches) performance on MLE-Bench.
Clear empirical gains from each component (knowledge base, intra-/cross-branch mechanisms), with demonstrated robustness across LLM backends (DeepSeek, Gemini, OpenAI). Also, paper is well writting and organised.

**Weaknesses:**

There is recent strong competing work, specifically MLE-STAR, which reports similar medal-rates

The novelty of MCGS over standard MCTS is mostly structural, mostly graph edge reuse.

Manual curation still has a large role, compromising the ability of this approach to scale, and potentially raising reproducibility questions

Evaluation limited to MLE-Bench; unclear generalization to non-Kaggle settings, unclear how much overfitting to some specific properties of Kaggle is happening.

No cost analysis for building and maintaining the knowledge base or for computational overhead of MCGS.

Limited interpretability analysis, how or when cross-branch references help remains opaque.

Paper does not address failure cases or negative results, leaving unclear when MCGS might degrade performance.

Lacks theoretical justification for why graph augmentation yields better convergence.

Paper would have been SOTA prior to MLEstar ...

**Questions:**

How is the graph structure in MCGS updated and pruned over time to prevent uncontrolled growth?
How does the agent decide when to trigger multi-branch aggregation versus normal expansion?
How large is the knowledge base, and how much manual effort was needed to curate and maintain it?
Could the knowledge base integration be automated or learned from prior runs?
What is the computational overhead of MCGS compared to MCTS (per iteration or per valid solution)?
How sensitive is performance to the choice of LLM backend?
What is the authors position regarding their approach compared to work such as MLEStar? 63% medals, 36%

---

> ### Author Response · Authors · 2025-11-23
> **Official Comment to Reviewer WoZe (Part 1 of 4)**
>
> > **[W1, W9, Q7] Discussion and comparison between our work and MLE-STAR.**
>
> We appreciate the reviewer for bringing attention to recent developments in the field and for pointing out this newly emerging competitive baseline. MLE-STAR [1] has indeed demonstrated strong performance within the MLE community, and we agree that it provides a meaningful contribution to automated ML engineering. In response to the reviewer’s question, we clarify the differences between the two approaches:
>
> 1. **Model capability differences and supplementary comparisons.** We further examined the latest version of MLE-STAR and found that its August 2025 update adopted the stronger Gemini-2.5-pro model, raising its MLE-Bench Lite medal rate from the earlier 43.9% to 63.6%. This number is indeed higher than the 62.1% reported in our paper using DeepSeek-R1. However, Gemini-2.5-pro incurs over **four times** the API cost of DeepSeek-R1 and typically generates substantially longer outputs. To ensure a fair comparison, we additionally evaluated a model comparable to 2.5-pro but significantly cheaper—**o4-mini**—which achieved **68.2%** medal rate on the same Lite benchmark, outperforming MLE-STAR despite using a lower-cost model.
>
> | Method          | Base model     | Running time(h) | Medal rate(%) |
> | --------------- | -------------- | --------------- | ------------- |
> | MLE-STAR        | Gemini-2.5-pro | 24              | 63.6          |
> | **InternAgent-MLE** | o4-mini        | **12**              | **68.2**          |
>
> 1. **Cost and efficiency comparison.** All our MLE-Bench experiments—both on the Full benchmark and the Lite benchmark—were conducted under a **12-hour** time limit, whereas MLE-STAR uses a default **24-hour** budget. Achieving comparable or even superior results with only half the time demonstrates that InternAgent-MLE offers a higher performance-to-cost ratio and stronger practicality in resource-constrained settings.
>
> Overall, we view the two approaches as complementary rather than directly competing: MLE-STAR emphasizes online retrieval of external knowledge and ensemble strategies, whereas InternAgent-MLE focuses on graph-structured search, cross-branch information flow, and operator-level control.
>
> [1] Nam, Jaehyun, et al. "MLE-STAR: Machine Learning Engineering Agent via Search and Targeted Refinement." *arXiv preprint arXiv:2506.15692* (2025).
>
> > **[W2] The novelty of MCGS over standard MCTS is mostly structural, mostly graph edge reuse.**
>
> Thank you for raising the question regarding the novelty of MCGS. We understand that the concern may come from viewing MCGS as mainly a structural modification of MCTS. However, within InternAgent-MLE, the role of MCGS goes well beyond a structural change:
>
> 1. **The graph models cross-branch information flow, not merely node reuse.** In standard MCTS, branches evolve independently and often fall into “local failure loops” on complex tasks. The three types of reference edges in MCGS—**intra-branch evolution**, **cross-branch reference**, and **multi-branch aggregation**—explicitly model how information should flow between branches during search. In MLE-Bench’s heterogeneous tasks, branches frequently fail for similar structural reasons due to limitations of the base LLM. MCGS enables the agent to recognize and incorporate patterns that succeed elsewhere, breaking isolation between branches and mitigating local‐optimum traps. This is fundamentally different from simply reusing past nodes.
> 2. **The graph works jointly with the operator system and knowledge base, forming a dynamic search flow.** Whether and how a reference edge is triggered depends on stagnation signals, global search status, and operator type. As search progresses, branch performance, stagnation length, and node outcomes continually update the agent’s decisions. This coupling between **structure, state, and operator policy** creates a form of *process-level learning*, where the search dynamically self-adjusts and self-evolves. Through this evolving flow, MCGS transitions smoothly between local refinement, cross-path integration, and global restructuring, improving feasibility, diversity, and convergence quality in large, non-convex code search spaces.
>
> In summary, MCGS is not a lightweight structural extension of MCTS, but a search-paradigm enhancement tailored to MLE-Bench that directly addresses cross-branch coordination and structural stagnation.

---

> ### Author Response · Authors · 2025-11-23
> **Official Comment to Reviewer WoZe (Part 2 of 4)**
>
> > **[W3, W5-1, Q3, Q4] The construction process, scale, labor cost, and scalability of the knowledge base.**
>
> We appreciate the reviewer’s questions regarding the construction, scale, level of human involvement, and extensibility of the knowledge base (KB). These are important concerns, and we clarify below that the KB is not manually curated but is built through a highly automated and reusable pipeline. Our response is structured in three parts.
>
> 1. **The KB is primarily constructed through automation rather than manual authoring.** InternAgent-MLE’s KB is generated through an automated pipeline consisting of three stages:
>    - **Automated extraction**：We automatically collect structured information from public model repositories (e.g., HuggingFace, GitHub), including typical model usage, input–output formats, and preprocessing patterns. We also extract widely adopted engineering strategies from open-source communities. These sources reflect general, task-agnostic engineering practices unrelated to any competition.
>    - **LLM-based cleaning**：An LLM performs automated cleaning, deduplication, formatting, and task-leakage filtering. It removes any content that could resemble or correspond to the 75 MLE-Bench tasks, including specific field names, pattern structures, or solution templates.
>    - **Final human review**：Human effort is limited to designing the pipeline and performing the final verification step for consistency and contamination checks.
>
> Thus, human involvement is minimal: content generation, cleaning, and structuring are all automated, while manual work is restricted to high-level supervision and final quality control.
>
> 2. **The KB is lightweight, requires minimal manual maintenance, and does not grow linearly with the benchmark.** The current KB contains only a few dozen model-level entries and several strategy-level entries, all built by a single PhD student once the pipeline was in place. Importantly, the KB consists of two categories of highly stable and task-agnostic knowledge:
>     - **Model-level knowledge:** common models and usage conventions in CV, NLP, Tabular, etc.
>     - **Data/strategy-level knowledge:** widely used engineering practices such as training stabilization techniques, data-cleaning steps, and default feature-processing patterns.
>
> Since these categories do not depend on the number of benchmark tasks, the KB does **not** grow proportionally with the size of MLE-Bench, and its long-term maintenance cost remains low. For transparency and reproducibility, we also include examples of KB entries in **Appendix A.1** of the revised manuscript.
>
> 3. **Regarding whether the KB can automatically learn from prior runs.** In the current implementation of InternAgent-MLE, KB updates occur **only within a single 12-hour setting.** During a run, the agent may temporarily store verified results and analyses in its memory and KB to guide subsequent steps within the same run. However, none of this information is written back to the persistent KB, nor accumulated across runs. This design strictly follows MLE-Bench’s rules and the 12-hour budget, **preventing any implicit use of experience from previous runs** that would violate the benchmark’s fairness. However, The search structure and KB mechanism inherently support cross-run continual learning: in unrestricted environments, the agent can write high-quality solutions to a persistent KB, automatically extract reusable patterns from past runs, and leverage them in future runs. This enables a process-level learning trajectory in which the system gradually accumulates expertise over long-horizon reasoning and large-scale search.
>
> > **[W5-2, Q5] What is the computational overhead of MCGS compared to MCTS?**
>
> According to the Reviewer's comment, we have supplemented  **in Page 25 of the revised manuscript.**
>
> Although MCGS introduces slightly longer inputs (more contextual information during expansion) per LLM call, **its overall cost does not increase**. Pure MCTS frequently produces low-quality candidates, triggering repeated LLM debug calls that never reach execution, leading to substantial wasted tokens.
> MCGS improves candidate success rates through cross-branch information flow, enabling more solutions to enter the execution phase—which is the dominant cost in MLE tasks. As a result, time is shifted from repeated LLM querying to actual execution, **reducing total LLM usage under the same 12-hour budget while achieving broader and more effective search coverage**.
>
> To further validate this, we compared the two methods across three representative tasks, and the table below reports the average input/output tokens and solution success rate. As shown, MCGS increases the success rate from 57.3% to 66.4% while reducing both input and output tokens.
>
> | **Method** | **# Input tokens** | **# Output tokens** | **Solution success rate** |
> | - | - | - | - |
> | InternAgent-MLE(MCTS) | 854K | 387K  | 57.3%  |
> | InternAgent-MLE(MCGS) | 720K | 282K | 66.4% |

---

> ### Author Response · Authors · 2025-11-23
> **Official Comment to Reviewer WoZe (Part 3 of 4)**
>
> > **[W4] Evaluation limited to MLE-Bench; unclear generalization to non-Kaggle settings, unclear how much overfitting to some specific properties of Kaggle is happening.**
>
> We appreciate the reviewer’s valuable comments regarding generalization. First, we clarify that MLE-Bench itself spans a wide range of tasks—including NLP, CV, tabular data, time series, and structured data—and our method does not rely on any Kaggle-specific templates, field names, or competition conventions during inference or search. Thus, the mechanism is inherently task-agnostic and is not tailored to any particular competition format.
>
> To further address the concern of whether the method works only for Kaggle-style tasks, we supplemented our evaluation with two tasks completely unrelated to Kaggle, both drawn from the **Autonomous Scientific Research** domain:
>
> - **AutoSenCls — Sentiment Analysis:** based on the SST-2 dataset [1], a classic binary sentiment classification task on movie reviews.
> - **Auto3DCls — 3D Point Cloud Classification:** evaluated on the ModelNet40 dataset [2], which contains 12,311 CAD object models across 40 common categories.
>
> We compare three frameworks:
> - **Baseline:** domain-specific baseline models (BERT-base [3] for AutoSenCls; PointNet [4] for Auto3DCls).
> - **InternAgent [5]:** a general-purpose automated scientific research framework composed of multiple specialized agents,  capable of autonomously designing research pipelines and iteratively refining them.
> - **InternAgent + MCGS:** replacing InternAgent’s original serial aiding-style code-generation agent with MCGS, turning linear search into graph-structured exploration.
>
> The results are shown below:
>
> | **Method** | **AutoSenCls (Acc)** | **Auto3DCls (OA)** |
> | - | - | - |
> | Baseline   | 91.0   | 91.0  |
> | InternAgent   | 93.5   | 95.5   |
> | InternAgent+ MCGS | **94.1**  | **96.5**|
>
> On both of these automated scientific tasks which are entirely different from Kaggle-style competitions, our framework consistently delivers additional and stable performance improvements.
>
> [1] Socher, Richard, et al. "Recursive deep models for semantic compositionality over a sentiment treebank." Proceedings of the 2013 conference on empirical methods in natural language processing. 2013.
>
> [3] Wu, Zhirong, et al. "3d shapenets: A deep representation for volumetric shapes." Proceedings of the IEEE conference on computer vision and pattern recognition. 2015.
>
> [4] Devlin, Jacob, et al. "Bert: Pre-training of deep bidirectional transformers for language understanding." Proceedings of the 2019 conference of the North American chapter of the association for computational linguistics: human language technologies, volume 1 (long and short papers). 2019.
>
> [5] Qi, Charles R., et al. "Pointnet: Deep learning on point sets for 3d classification and segmentation." Proceedings of the IEEE conference on computer vision and pattern recognition. 2017.
>
> [5] Team, InternAgent, et al. "InternAgent: When Agent Becomes the Scientist--Building Closed-Loop System from Hypothesis to Verification." arXiv e-prints (2025): arXiv-2505.
>
> > **[W6, W8] Interpretability of Cross-Branch References and Benefits of Graph Augmentation**
>
> We fully appreciate the reviewer’s interest in the cross-branch reference mechanism and the role of the graph structure. To clarify these points, we provide a more explicit explanation here and have added visual examples in the revised manuscript(Appendix A.6).
>
> First, the triggering of **cross-branch reference** is fully interpretable. It is governed by the progress of the current branch and the agent’s internal assessment, with the most important signal being *local branch stagnation*. When a branch has accumulated enough feasible nodes but none of the successful nodes improve upon its historical best over several consecutive iterations, we treat it as stagnant. In such cases, further local perturbations are unlikely to produce gains, so the agent triggers cross-branch reference to incorporate higher-quality structural information from other branches.
>
> Second, regarding the **effectiveness and interpretability of the graph structure**, we have added multiple visualizations in **Appendix A.6**. These examples (**Fig. 12,13,14**) illustrate how the agent inspects information from other branches, how reference edges guide its design choices, and how structural fusion is performed in practice. The cases provide an intuitive view of how information flows through the graph and demonstrate how MCGS leverages structured cross-branch signals to guide the search more steadily toward convergence.

---

> ### Author Response · Authors · 2025-11-23
> **Official Comment to Reviewer WoZe (Part 4 of 4)**
>
> > **[W7] Paper does not address failure cases or negative results, leaving unclear when MCGS might degrade performance.**
>
> We appreciate the reviewer’s concern about MCGS performance in failure cases and have supplemented  in **Page 9** of the revised manuscript. Indeed, MCGS does not guarantee improvements under all task conditions. In our observations, performance drops are mainly due to the limitations of the underlying base model rather than any inherent weakness of the graph structure.
>
> In some high-difficulty tasks, feasible nodes are extremely rare, and multiple branches may repeatedly produce syntactic or logical errors. In such cases, cross-branch reference cannot obtain any reusable high-quality structure. When all branches lack strong candidates, fusion naturally has little useful information to combine and may occasionally introduce extra complexity.
>
> | **Task**                        | Bronze medal score | InternAgent-MLE(MCTS) | InternAgent-MLE(MCGS) |
> | ------------------------------- | ------------------ | --------------------- | --------------------- |
> | bms-molecular-translation(↓)    | 1.99               | 73.19                 | 91.1853               |
> | nfl-player-contact-detection(↑) | 0.69652            | 0.55136               | 0.51394               |
>
> In future work, we plan to improve the base model’s code-generation reliability and to adaptively reduce the strength of graph-based fusion in extremely challenging tasks, helping mitigate these failure modes more effectively.
>
> > **[Q1] How is the graph structure in MCGS updated and pruned over time to prevent uncontrolled growth?**
>
> Although MCGS introduces a graph to enable cross-branch infor mation flow, its size does not expand uncontrollably. Node creation and edge generation are tightly constrained by operator-trigger conditions, stagnation signals, and the time budget—not by unrestricted expansion. More importantly, in MLE tasks the dominant cost is code execution and model training rather than graph updates. Under a fixed 12-hour budget, the number of nodes that can be generated is naturally limited by execution time, preventing uncontrolled growth of the graph. Therefore, in our setting, **MCGS does not require additional pruning mechanisms** to maintain controllable scale.
>
> > **[Q2] How does the agent decide when to trigger multi-branch aggregation versus normal expansion?**
>
> In InternAgent-MLE, beyond standard expansion, the reference-edge operations include intra-branch evolution, cross-branch reference, and multi-branch aggregation. The first two are primarily activated when the current branch becomes *stagnant*—i.e., it fails to improve its historical best over several iterations—and are used to strengthen that branch. In contrast, multi-branch aggregation serves a different purpose: it aims to restart a new branch from the root by combining structural signals from multiple strong branches. This operation is triggered only when three conditions hold simultaneously: **(1)** the search has entered the mid-to-late stage of the time budget, ensuring sufficient exploration depth; **(2)** at least two branches have produced stable, error-free, high-performing nodes that provide a meaningful basis for aggregation; and **(3)** the global best solution has not improved for an extended period, indicating diminishing returns from single-branch refinement. Under these conditions, aggregation can generate a new, structurally advantageous trajectory from the root and help the agent escape local optima.
>
> > **[Q6] How sensitive is performance to the choice of LLM backend?**
>
> We appreciate the reviewer’s question regarding model compatibility. MCGS is not coupled to any specific LLM backend—its core mechanism depends on the search structure and cross-branch information flow rather than model-specific internal capabilities. In principle, MCGS can be combined with any LLM that supports code generation. As shown in Figure 5(a) of the paper, we evaluated InternAgent-MLE with multiple backends, including DeepSeek, OpenAI, and Gemini, using identical parameter settings. The results demonstrate that InternAgent-MLE operates consistently across different models and improves proportionally as the underlying LLM becomes stronger. Overall, InternAgent-MLE is a model-agnostic framework with **low sensitivity to the choice of LLM backend**, providing strong portability and stability.

---

> > ### Comment · Reviewer_WoZe · 2025-11-25
> >
> > I appreciate the thorough response to each of my concerns. Although I remain of the opinion that given the performance of competitive approaches (even if less cost efficient) this is not a must-have contribution to ICLR, especially the points regarding the changes via MCGS being more than "just" structural are well taken. Although I am not fully convinced in light of how crowded the "medal"-benchmark is getting, and how quickly the SOTA there is rising, I am adjusting my rating to marginally above the acceptance threshold, without raising my confidence level.

---

> > > ### Author Response · Authors · 2025-11-26
> > > **Official Comment by Authors**
> > >
> > > We sincerely thank the reviewer for the thoughtful response and for reconsidering the novelty of the MCGS framework. We appreciate that you have taken the time to review our rebuttal and adjust your assessment.
> > >
> > > We fully understand the concern regarding the rapidly evolving and competitive nature of the MLE benchmark. However, we believe that as leaderboard performance approaches saturation, efficiency and cost-effectiveness become the critical differentiators for practical adoption. By achieving competitive results with significantly lower computational costs than recent SOTA approaches, InternAgent-MLE offers a necessary perspective on efficient agentic search.
> > >
> > > We appreciate your constructive feedback, which has helped us further strengthen the work, and we will continue improving the performance and generality of the framework in future research.

---

### Official Review · Reviewer_yhRj · 2025-10-28

**Soundness:** 3
**Presentation:** 3
**Contribution:** 2
**Rating:** 6
**Confidence:** 4

**Summary:**

This paper introduces InternAgent-MLE, an LLM-based coding agent designed for Machine Learning Engineering (MLE) tasks. The key contributions are: (1) a curated ML domain knowledge base providing task-specific priors, and (2) Monte Carlo Graph Search (MCGS), which extends MCTS by introducing graph edges for cross-branch knowledge sharing and solution fusion. Evaluated on MLE-Bench, the method achieves 36.4% average medal rate under a 12-hour budget, outperforming existing baselines.

**Strengths:**

1. The paper clearly identifies limitations of existing tree-based search methods, specifically node isolation and limited knowledge transfer across branches. The motivation for graph-based search is compelling.
2. MCGS extends MCTS in a clear way by introducing reference edges (E_ref) while maintaining tree edges (E_T) for backpropagation. The four expansion modes (primary, intra-branch evolution, cross-branch reference, multi-branch aggregation) provide systematic ways to leverage historical information. The approach is well motivated and sound.
3. The method achieves state-of-the-art performance with 36.4% average medal rate at the submission time (as of today, the SOTA is 43.5%) proving its efficacy.
4. The paper includes ablations demonstrating the value of each component, analysis across different LLMs, and temporal analysis showing progressive improvement.
5. The paper is generally well written and easy to follow.

**Weaknesses:**

1. The primary technical contribution, Monte Carlo Graph Search, is presented as a novel variant of MCTS. However, the idea of augmenting MCTS with graph structures to share information, merge nodes, or reuse trajectories has been explored in various forms in the search and planning literature. The related work section focuses heavily on other MLE agents but lacks a discussion of how MCGS differs from or builds upon other graph-based MCTS variants.
2. The curated knowledge base (KB) is a critical component, responsible for a ~9% absolute improvement in the medal rate according to the ablation study. However, its construction is described vaguely as "synthesizing practices from open-source repositories... followed by careful selection". The engineering effort and generalizability of this component are unclear. The paper would benefit from sharing at least all the prompts and pseudocode for building such a knowledge base. Moreover, the paper does not address potential data contamination –
there is no discussion of whether the knowledge base includes competition-specific heuristics that could appear from different sources.
3. Critical design decisions lack clear specification. For example:
  - When is "cross-branch reference" triggered? What constitutes a "stagnant" branch?
  - How is "multi-branch aggregation" triggered? The phrase "heuristically spawned" is vague.
4. Some ablations are missing but desirable, e.g., impact of knowledge base design choices (model-level vs. data-level vs. strategy-level)

**Questions:**

1. Could authors detail the KB decontamination: which repositories/solutions were included/excluded? Is there any overlap with the 75 tasks? Could you provide the examples how the KB looks like? Is the construction of KB an automated process or does it require manual curation?
2. What are the precise triggering conditions for intra-branch, cross-branch and multi-branch operations?
3. Could authors provide a reward structure, i.e. how different actions are rewarded/penalized?
4. How sensitive is performance to the hyperparameters in Table 4 (e.g., branch_top_k, max_ref_num)? What are the most important ones? Were these tuned on the test set?

---

> ### Author Response · Authors · 2025-11-23
> **Official Comment to Reviewer yhRj (Part 1 of 3)**
>
> > **[W1] The related work section focuses heavily on other MLE agents but lacks a discussion of how MCGS differs from or builds upon other graph-based MCTS variants.**
>
> We acknowledge the reviewer’s suggestion and have added **Section 2.3** in the revised Related Work section.
>
> First, graph-based MCTS methods in planning and RL [1] [2] assume that states can be judged similar or merged, allowing the graph to reduce redundant expansion. This assumption does not hold in MLE settings: each node is an LLM-generated code solution, structurally complex and not mutually equivalent, making state merging infeasible.
> Second, LLM-based coding-agentssuch as LocAgent [3] and CodexGraph [4] employ graphs as static representations (e.g., dependency graphs or code-base relations) for retrieval or localization. These graphs do not participate in generative search, nor do they evolve with node expansion, unlike the **dynamic search graph** in MCGS.
> Third, graph-based reasoning works [5] model external relational graph structures and apply MCTS. In contrast, MCGS uses a search graph to represent the search process itself, enabling cross-path references and structural composition, addressing the key limitation of traditional MCTS in MLE.
>
> [1] Czech, Johannes, Patrick Korus, and Kristian Kersting. "Monte-Carlo graph search for AlphaZero." *arXiv preprint arXiv:2012.11045* (2020).
>
> [2] Leurent, Edouard, and Odalric-Ambrym Maillard. "Monte-carlo graph search: the value of merging similar states." *Asian Conference on Machine Learning*. PMLR, 2020.
>
> [3] Chen, Zhaoling, et al. "Locagent: Graph-guided llm agents for code localization." *Proceedings of the 63rd Annual Meeting of the Association for Computational Linguistics (Volume 1: Long Papers)*. 2025.
>
> [4] Liu, Xiangyan, et al. "Codexgraph: Bridging large language models and code repositories via code graph databases." *Proceedings of the 2025 Conference of the Nations of the Americas Chapter of the Association for Computational Linguistics: Human Language Technologies (Volume 1: Long Papers)*. 2025.
>
> [5] Ma, Yingwei, et al. "How to understand whole software repository." *arXiv preprint arXiv:2406.01422* (2024).
>
> >**[W3]  What constitutes a "stagnant" branch? What are the precise triggering conditions for intra-branch, cross-branch and multi-branch operations?  The phrase "heuristically spawned" is vague.**
>
> We appreciate the reviewer’s suggestion to further clarify the triggering mechanism and have added a more explicit description in **Page 6 of the revised manuscript**. Here, we briefly address the reviewer’s concern from the three requested perspectives:
>
> 1. **Definition of a “stagnant branch.”** We consider a branch to be *stagnant* when it has already produced a sufficient number of valid nodes but, over several consecutive iterations, none of the successful nodes achieve a metric that surpasses the branch’s historical best. In this situation, continuing fine-grained local perturbations is unlikely to yield improvements, and this stagnation signal becomes the key condition for triggering a larger-step exploration.
>
>  2. **Triggering conditions of the three expansion operations.** These operations are not triggered arbitrarily; their activation depends on branch stagnation, other branches’ progress, and execution feedback (case examples in Appendix A.2 and A.6).
> - **Intra-branch evolution.** When a branch becomes stagnant—i.e., repeated iterations fail to improve its historical best—the probability of triggering larger structural modifications increases, allowing the agent to escape local optima.
> - **Cross-branch reference.** If a branch is stagnant while other branches have already produced stronger solutions, the system increases the chance of referencing those branches, enabling the current path to incorporate their key structures instead of continuing ineffective local refinements.
> - **Multi-branch aggregation.** This operation is limited to the later stage of search and is triggered only when: the search has entered the mid-to-late budget, multiple branches have already yielded stable and complementary high-quality nodes, and the global best has not improved for a prolonged period. When these conditions align, aggregation may be activated to create a fused candidate from the root and help break global stagnation.
>
> 3. **Meaning of “heuristically spawned.”** We apologize for the confusion caused by our earlier wording and revise it in the manuscript. The term does not refer to random or opaque behavior.  Instead, it denotes a probabilistic scheduling mechanism based on observable signals such as stagnation duration, the number of successful branches, the agent’s internal reasoning, and node execution feedback.

---

> ### Author Response · Authors · 2025-11-23
> **Official Comment to Reviewer yhRj (Part 2 of 3)**
>
> >  **[W2 ,Q1] Knowledge Base: Construction, Decontamination, and Transparency**
>
> **[W2-1, W2-2, Q1-3, Q1-4] The engineering effort and generalizability of this component are unclear. Could you provide the examples how the KB looks like? Is the construction of KB an automated process or does it require manual curation?**
>
> We fully appreciate the reviewer’s attention to the construction of the knowledge base (KB). For transparency and reproducibility, we have added concrete KB examples in **Appendix A.1** of the revised manuscript.
>
> The KB in InternAgent-MLE is not built through extensive manual curation but is generated through an **automated pipeline**. The pipeline first extracts structured information from common open-source model repositories (e.g., HuggingFace, GitHub), including typical model usages, input–output formats, and recommended preprocessing steps. It then collects widely adopted, task-agnostic engineering practices from open-source communities—such as data augmentation, pseudo-labeling, and model ensembling—which represent broadly accepted strategie.
>
> After gathering the raw content, we use an LLM to automatically clean, deduplicate, and screen it for potential task-leakage risks, ensuring that the KB contains no information tied to any of the 75 MLE-Bench tasks. Human review is only required for final consistency checks and leakage verification.
>
> Importantly, the KB does not scale linearly with the size of the benchmark because it primarily contains two types of stable, task-independent knowledge: (1) model-level common practices across CV, NLP, Tabular tasks, and (2) data/strategy-level general engineering strategies. These categories are **highly reusable** and remain valid even as the benchmark evolves. We also plan to extend the pipeline to automate the collection of more specialized domain knowledge, such as in the scientific field, to enhance the knowledge supplementation of the framework, enabling long-term KB maintenance.
>
> **[W2-5, Q1-1, Q1-2] The paper does not address potential data contamination. Which repositories/solutions were included/excluded? Is there any overlap with the 75 tasks?**
>
> Thank you for raising the concern regarding potential contamination and task overlap in the knowledge base (KB). We clarify the KB’s sources, filtering procedures, and decontamination mechanisms. InternAgent-MLE strictly follows a task-agnostic construction principle, with all KB content generated through automated pipelines and subjected to multi-stage leakage checks to ensure there is no overlap with the 75 MLE-Bench tasks.
>
> Regarding data sources, repositories/solutions are included only from **general, non-competition** public resources, such as:
>
> - HuggingFace model repositories
>   - https://huggingface.co/models?pipeline_tag=image-classification
>   - https://huggingface.co/models?pipeline_tag=text-classification
>   - https://huggingface.co/models?pipeline_tag=translation
>   - ...
> - Generic ML engineering example repositories
>   - https://github.com/xxxx/100-Days-Of-ML-Code
>   - ...
>
> These sources contain widely used model descriptions, preprocessing conventions, and general engineering practices, but no task-specific solutions.
>
> To prevent leakage, we explicitly exclude all repositories that may **contain Kaggle-specific solutions or code**, such as:
> - https://github.com/xxxx/kaggle-solutions
> - https://github.com/XXX/kaggle-aptos2019-blindness-detection
> - ...
>
> We apply a strict decontamination pipeline to ensure that the KB contains no content related to any MLE-Bench task. First, automated scripts perform keyword-level filtering: any extracted text containing names, fields, path patterns, or dataset identifiers associated with the 75 tasks are immediately removed. Second, we use LLM for semantic-level screening to detect and exclude content structurally similar to Kaggle-related solutions. After these automated checks, we conduct a final manual review to verify formatting and confirm that the KB is free from contamination. Through this multi-layer filtering process, we ensure that the KB is fully independent of 75 MLE-Bench tasks.
>
> > **[W4] Some ablations are missing but desirable, e.g., impact of knowledge base design choices (model-level vs. data-level vs. strategy-level)**
>
> We appreciate the reviewer’s suggestion regarding whether different components of the knowledge base contribute meaningfully. To make this clearer, we added new ablation experiments evaluating the three categories of knowledge, provided in **Appendix A.5**.
>
> | Knowledge Base Design  | learning-agency-lab-automated-essay-scoring-2 (↑) | stanford-covid-vaccine (↓) |
> | - | - | - |
> | w/o knowledge  | 0.8109 |0.27584|
> | + model-level knowledge | 0.82921 | 0.2379|
> | + model & data-level knowledge |0.83304 |0.23225|
> | **InternAgent-MLE (full KB)**|**0.84965**|**0.22527**|
>
> The results show that each component offers significant and complementary benefits, confirming that the KB design is not redundant.

---

> ### Author Response · Authors · 2025-11-23
> **Official Comment to Reviewer yhRj (Part 3 of 3)**
>
> > **[Q3] Could authors provide a reward structure, i.e. how different actions are rewarded/penalized?**
>
> Thanks for raising this important question. We have added a clearer definition and formal description of the reward function **in Page 6-7 of the revised manuscript** to improve transparency and reproducibility.
>
> Importantly, in InternAgent-MLE, the reward does **not** depend on the operator type; it depends solely on whether the node executes successfully and whether it provides actual performance improvement. Since MLE-Bench evaluates the final metric, assigning different rewards to different operators would introduce unintended bias and reduce comparability. Thus, the reward is designed around execution validity and performance contribution, rather than operator categories.
>
> Specifically, we formally define the immediate reward $R(v)$ as:
> $$
> R(v) =
> \begin{cases}
> -1, & \text{if execution fails or no valid metric is obtained}, \\\\
> 1,  & \text{if execution succeeds but does not improve the branch best}, \\\\
> 2,  & \text{if execution succeeds and refreshes the branch best metric}.
> \end{cases}
> $$
>
> This design keeps the reward function simple while clearly distinguishing failed attempts, feasible attempts, and genuine improvements, making credit assignments more interpretable.
>
> > **[Q4] How sensitive is performance to the hyperparameters in Table 4 (e.g., branch_top_k, max_ref_num)? What are the most important ones? Were these tuned on the test set?**
>
> We would like to first clarify that none of the hyperparameters were tuned on the test set; all settings were chosen based on the public training data and development-phase experience to ensure fair evaluation.
>
> Overall, the method is not highly sensitive to hyperparameters. All hyperparameters were set within empirically reasonable ranges based on development experience. Small-range perturbations during development showed only limited variation, consistent with our design: **the main gains come from the search structure rather than tuning any single parameter.** Among the parameters, those controlling cross-branch information flow such as max_ref_num and max_agg_num have a more noticeable effect, while the UCT constant or max_parallel_workers primarily affect search efficiency and have minor impact on final performance.
>
> To validate stability, we added sensitivity experiments for branch_top_N and max_ref_num in **Appendix A.9.**
>
> | branch_top_N | random-acts-of-pizza (↑) | stanford-covid-vaccine (↓) |
> | ------------ | ------------------------ | -------------------------- |
> | 1            | 0.6381                   | 0.3056                     |
> | 5            | **0.7298**                   | **0.2253**                    |
> | 10           | 0.7286                   | 0.2465                     |
>
> | max_ref_num | random-acts-of-pizza (↑) | stanford-covid-vaccine (↓) |
> | ----------- | ------------------------ | -------------------------- |
> | 1           | 0.5725                   | 0.3490                     |
> | 5           | 0.7222                   | 0.2465                     |
> | 7           | **0.7298**                   | **0.2253**                     |
> | 9           | 0.7109                   | 0.2473                     |
>
> The results show that performance remains stable within reasonable ranges; only excessively large values introduce noise, and overly small values weaken cross-branch information flow.

---

### Official Review · Reviewer_tcRV · 2025-10-31

**Soundness:** 2
**Presentation:** 2
**Contribution:** 2
**Rating:** 4
**Confidence:** 3

**Summary:**

This paper proposes **InternAgent-MLE**, a coding agent designed for end-to-end Machine Learning Engineering (MLE) tasks such as Kaggle competitions. The framework integrates three core components:
1. **A curated ML knowledge base**providing model-, data-, and strategy-level priors to reduce cold-start errors of initial solutions;
2. **A Monte Carlo Graph Search (MCGS) algorithm**, extending Monte Carlo Tree Search (MCTS) by allowing nodes to incorporate information from non-parent nodes during expansion—thereby enabling trajectory recall, cross-branch reference, and multi-branch aggregation.
3. **A fine-grained operator set** that stabilizes search and improves executability.

InternAgent-MLE achieves state-of-the-art performance on **MLE-Bench**, reaching a 36.4% average medal rate and 18.7% gold medal within a 12-hour runtime—half the standard evaluation time.

**Strengths:**

1. The proposed MCGS mechanism allows efficient reuse of previous solutions both within branches or across branches.
2. The agent demonstrates strong empirical performance, outperforming all existing baselines on MLE-Bench in both efficiency and robustness.

**Weaknesses:**

1. The paper uses the term 'Monte Carlo Graph Search (MCGS)' without referencing prior work under the same name. Existing literature—such as [1] and [2]—already explores graph extensions of MCTS. The authors should clarify whether their formulation is conceptually novel and cite these precedents to avoid ambiguity.
2. The operator selection policy remains under-specified. It is unclear how the agent decides which operator to apply.
3. Details about reward propagation could be elaborated—particularly how rewards are updated across parent nodes.

**References**

[1] Czech, Johannes, Patrick Korus, and Kristian Kersting. 'Monte-Carlo graph search for AlphaZero.' arXiv preprint arXiv:2012.11045 (2020).

[2] Leurent, Edouard, and Odalric-Ambrym Maillard. 'Monte-carlo graph search: the value of merging similar states.' Asian Conference on Machine Learning. PMLR, 2020.

**Questions:**

1. **Novelty clarification:** Has MCGS been attempted in other coding-agent contexts besides MLE? Or more generally, has a similar graph-structured search been attempted in other coding-agent contexts besides MLE?
2. **Graph structure:** In Figure 2 and Eq. (8), should the new node created by multi-branch aggregation indeed be connected to the root node to somehow denote it as a starting node.
3. **Operator selection:** How is the next operator chosen?
4. **Reward updates:** During backpropagation, how do all ancestors along the trajectory update their reward values?
5. **Notation consistency:** In Line 293, 'graph-level top-$k$' seems to correspond to the 'top-$N$' reference set in Eq. (7); aligning the notation would improve clarity.
6. **Edge usage:** Since $E_{ref}$ is excluded from backpropagation, is it still necessary to keep this recorded?

---

> ### Author Response · Authors · 2025-11-23
> **Official Comment to Reviewer tcRV (Part 1 of 3)**
>
> > **[W1] Existing literature such as [1] and [2] already explores graph extensions of MCTS. The authors should clarify whether their formulation is conceptually novel and cite these precedents to avoid ambiguity.**
>
> We sincerely appreciate the reviewer’s comment. In the revised version, we have added citations to prior MCGS work [1] [2] in **Section 2.3** and clarified how our method differs from these formulations.
>
> Both prior works [1] [2] are designed for **planning or game-playing settings with well-defined state spaces**. Their graph structure is primarily used to **merge similar states** in order to reduce redundant exploration and repeated sampling, requiring tailored modifications to selection and backpropagation to handle multiple parent nodes.
> In contrast, InternAgent-MLE targets an **open-ended LLM code-generation scenario**. Each node corresponds to a generated code candidate, and the objective is to broaden the search space within a limited time budget. Since there is no well-defined state space, traditional state merging is unnecessary. Our graph structure is introduced not to eliminate redundancy but to **mitigate issues such as isolated branches and limited information reuse**. The reference edges (intra-branch evolution, cross-branch reference, and multi-branch aggregation) facilitate richer information flow during expansion without additional computation for multi-parent backpropagation.
>
> We have referenced prior work in the revised manuscript. We appreciate the reviewer’s suggestion, which helps delineate conceptual boundaries more clearly and highlight the contributions of our approach.
>
> [1] Czech, Johannes, Patrick Korus, and Kristian Kersting. "Monte-Carlo graph search for AlphaZero." arXiv preprint arXiv:2012.11045 (2020).
>
> [2] Leurent, Edouard, and Odalric-Ambrym Maillard. "Monte-Carlo graph search: the value of merging similar states." Asian Conference on Machine Learning. PMLR, 2020.
>
> > **[W2, Q3] The operator selection policy remains under-specified. It is unclear how the agent decides which operator to apply.**
>
> We fully appreciate the reviewer’s comment. In the revised version, we provide a clearer and more concise explanation of the operator-selection mechanism. Overall, operator choice depends on the node’s current state and the progress of the search.
>
> At the start of the search, when no runnable solution exists, the agent uses the **draft** operator to generate initial candidates. If execution errors occur, the **debug** operator is triggered to revise the code based on the error message until the issue is fixed or the retry limit is reached. Once a node runs successfully and produces a valid metric, the agent switches to the optimization stage and invokes the **improve** operator. Within this phase, **Improve-FE** and **Improve-CS** operators are sampled probabilistically to maintain exploration while using the knowledge base for heuristic guidance.
>
> For graph-related **fusion** operators (the three reference-edge types in Section 3.3.2), intra-branch evolution and cross-branch reference are triggered when a branch stagnates after several rounds of failed improvement. Multi-branch aggregation is activated only under stricter conditions—typically in the latter half of the time budget, when multiple branches already contain high-quality solutions and the global best metric has not improved for a long period—at which point it merges information and starts a new trajectory from the root.
>
> In addition, every newly generated solution is checked using the **code review** operator, and near the end of the time budget, the **ensemble** operator aggregates all promising candidates to fully utilize the accumulated progress.
>
> According to the Reviewer's comment, we have supplemented  in **Page 6-7** of the revised manuscript.
>
> > **[W3, Q4] Details about reward propagation. How rewards are updated across parent nodes?**
>
> Thank you for highlighting that our explanation of the reward backpropagation mechanism was not sufficiently clear. In the revised version, we have added more explicit definitions of the reward values and a clearer description of the backpropagation process in **section 3.3.2**.
>
> In InternAgent-MLE, each node receives an immediate reward $R(v)$ after execution. We formally define the immediate reward $R_v$ as:
>
> $$
> R(v) =
> \begin{cases}
> -1, & \text{if execution fails or no valid metric is obtained}, \\\\
> 1,  & \text{if execution succeeds but does not improve the branch best}, \\\\
> 2,  & \text{if execution succeeds and refreshes the branch best metric}.
> \end{cases}
> $$
>
> The reward is then backpropagated along the main search path from node $v$ to the root. Each ancestor node $u$ updates its visit count and cumulative reward following the standard MCTS rules:
>
> $$N_u \leftarrow N_u + 1,\qquad W_u \leftarrow W_u + R(v),$$
>
> and obtains the new mean value:
>
> $$Q_u = \frac{W_u}{N_u}.$$

---

> ### Author Response · Authors · 2025-11-23
> **Official Comment to Reviewer tcRV (Part 2 of 3)**
>
> > **[Q1] Novelty clarification**
>
> **[Q1-1] Has MCGS been attempted in other coding-agent contexts besides MLE?**
>
> To verify whether MCGS exhibits cross-task generalization ability, we conduct two experiments in the Autonomous Scientific Research domain  entirely unrelated to MLE, built on the representative work InternAgent [1] :
>
> - **AutoSenCls — Sentiment Analysis:** based on the SST-2 dataset [2], a classic binary sentiment classification task on movie reviews.
> - **Auto3DCls — 3D Point Cloud Classification:** evaluated on the ModelNet40 dataset [3], which contains 12,311 CAD object models across 40 common categories.
>
> We compare three frameworks:
> - **Baseline:** domain-specific baseline models (BERT-base [4] for AutoSenCls; PointNet [5] for Auto3DCls).
> - **InternAgent:** a general-purpose automated scientific research framework composed of multiple specialized agents, capable of autonomously designing modeling pipelines and iteratively refining them.
> - **InternAgent + MCGS:** replacing InternAgent’s original serial aiding-style code-generation agent with MCGS, turning linear search into graph-structured exploration.
>
> The results are shown below:
>
> | **Method**        | **AutoSenCls (****Acc****)** | **Auto3DCls (****OA****)** |
> | -| - | - |
> | Baseline          | 91.0                         | 91.0                       |
> | InternAgent       | 93.5                         | 95.5                       |
> | InternAgent+ MCGS | **94.1**                     | **96.5**                   |
>
> We observe that on both automated scientific tasks which are entirely different from MLE-Bench, MCGS consistently brings additional and stable performance improvements. This indicates that MCGS does not rely on any specific task structure; instead, it serves as a generally effective mechanism for enhancing code search and optimization.
>
> [1] Team, InternAgent, et al. "InternAgent: When Agent Becomes the Scientist--Building Closed-Loop System from Hypothesis to Verification." *arXiv e-prints* (2025): arXiv-2505.
>
> [2] Socher, Richard, et al. "Recursive deep models for semantic compositionality over a sentiment treebank." *Proceedings of the 2013 conference on empirical methods in* *natural language processing*. 2013.
>
> [3] Wu, Zhirong, et al. "3d shapenets: A deep representation for volumetric shapes." *Proceedings of the IEEE conference on* *computer vision and pattern recognition*. 2015.
>
> [4] Devlin, Jacob, et al. "Bert: Pre-training of deep bidirectional transformers for language understanding." *Proceedings of the 2019 conference of the North American chapter of the association for computational linguistics: human language technologies, volume 1 (long and short papers)*. 2019.
>
> [5] Qi, Charles R., et al. "Pointnet: Deep learning on point sets for 3d classification and segmentation." *Proceedings of the IEEE conference on* *computer vision and pattern recognition*. 2017.
>
> **[Q1-2] Has a similar graph-structured search been attempted in other coding-agent contexts besides MLE?**
>
> In the coding-agent domain, several methods indeed make use of graph structures, such as LocAgent [6], VerilogCoder [7], and CodexGraph [8], which rely on code dependency graphs, task–circuit relation graphs, or graph databases for localization, planning, or context retrieval. However, these graphs are used for **static structural representation** or information lookup; they do not participate in the generative search process, nor do they evolve with node expansion or allow information to flow across different search paths.
>
> In contrast, InternAgent-MLE incorporates graph structures directly into the **dynamic search process**: each node represents an executable candidate, and the graph continuously updates as new candidates are generated. Cross-branch references and multi-branch aggregation actively influence the search trajectory and the final solution. MCGS combines the flexibility of graph-based information flow with the stable backpropagation mechanism of tree search, targeting open-ended code generation. We have added citations to prior work in **Section 2.3** and clarified how our method differs from these formulations.
>
> [6] Chen, Zhaoling, et al. "Locagent: Graph-guided llm agents for code localization." *Proceedings of the 63rd Annual Meeting of the Association for Computational Linguistics (Volume 1: Long Papers)*. 2025.
>
> [7] Ho, Chia-Tung, Haoxing Ren, and Brucek Khailany. "Verilogcoder: Autonomous verilog coding agents with graph-based planning and abstract syntax tree (ast)-based waveform tracing tool." *Proceedings of the AAAI Conference on* *Artificial Intelligence*. Vol. 39. No. 1. 2025.
>
> [8] Liu, Xiangyan, et al. "Codexgraph: Bridging large language models and code repositories via code graph databases." *Proceedings of the 2025 Conference of the Nations of the Americas Chapter of the Association for Computational Linguistics: Human Language Technologies (Volume 1: Long Papers)*. 2025.

---

> ### Author Response · Authors · 2025-11-23
> **Official Comment to Reviewer tcRV (Part 3 of 3)**
>
> > **[Q2] In Figure 2 and Eq. (8), should the new node created by multi-branch aggregation indeed be connected to the root node to somehow denote it as a starting node.**
>
> We genuinely appreciate the reviewer’s careful observation regarding the graph-structure detail. You are absolutely correct: according to the definition in Equation (8), a newly generated node from multi-branch aggregation should indeed start a new trajectory from the root, and thus should be structurally connected to the root node. We apologize for this oversight and have corrected **Figure 2** in the revised manuscript to ensure full consistency between the illustration and the actual algorithmic process.
>
> > **[Q5] In Line 293, 'graph-level top-k' seems to correspond to the 'top-N' reference set in Eq. (7); aligning the notation would improve clarity.**
>
> We are grateful for the reviewer’s observation regarding the inconsistent use of symbols. We have unified all related notation to “top-N” in the revised manuscript and clarified the scope of its usage to avoid any confusion. The terms “top-N” and “top-k” indeed refer to the same concept—selecting the set of the top N best nodes. The only difference in the manuscript was contextual emphasis: Equation (7) focuses on the best-node sets used during cross-branch referencing, whereas the description at line 293 highlights the need to retain several top nodes at both the branch and graph levels as memory storage.
>
> > **[Q6] Since $E_{ref}$ is excluded from backpropagation, is it still necessary to keep this recorded?**
>
> We appreciate the reviewer’s question regarding the necessity of keeping $E_{\text{ref}}$. Although $E_{\text{ref}}$ does not participate in reward backpropagation, it remains essential to record for MCGS.
>
> First, $E_{\text{ref}}$ records the dependency created by fusion-related expansions. These cross-branch or aggregated relationships cannot be represented by the tree edges $E_T$, and without $E_{\text{ref}}$ the algorithm would lose the ability to trace where fused nodes draw their information from.
> Second, $E_{\text{ref}}$ is important for interpretability and post-hoc analysis, allowing us to inspect which branches contribute to a fused trajectory and how information is reused across paths.
> Finally, maintaining $E_{\text{ref}}$ is nearly cost-free: it only stores index references during expansion and does not introduce additional computation or memory overhead.
>
> For these reasons, even though $E_{\text{ref}}$ is not used for value backpropagation, it is still necessary for preserving information flow, structural correctness, and interpretability in MCGS.

---

### Official Review · Reviewer_bkGv · 2025-11-01

**Soundness:** 4
**Presentation:** 3
**Contribution:** 3
**Rating:** 6
**Confidence:** 3

**Summary:**

This article introduces a novel large language model (LLM) agent, **InternAgent-MLE**, designed to enhance the **stability and convergence speed** of LLMs in handling **Machine Learning Engineering (MLE**) tasks. Specifically, the paper builds upon the existing **Monte Carlo Tree Search (MCTS**) framework and proposes the use of **Monte Carlo Graph Search (MCGS**) to optimize the agent's ability to leverage **historical information**. Additionally, it establishes a connection between a curated knowledge base and the LLM to improve **warm-start efficiency**, and employs **parallel exploration** to accelerate convergence.
The article provides a deep analysis of the limitations of current LLM agents utilizing MCTS. It focuses on the **isolation of information within different branches** as a key motivator for improving the agent’s capabilities. The main body of the paper introduces the search logic of MCGS and the naming of different types of nodes based on their functions. It then discusses the operational mechanisms of MCGS, including the enablement and updating of **primary and reference edges**, around Figure 2.
In the experimental section, the authors use **MLE-Bench** and **MLE-Bench-lite** as evaluation datasets and compare the performance of various methods with InternAgent-MLE in terms of **medal rates and times to Medal at different granularity levels**. They also compare the **beat ratio** between InternAgent-MLE and the baseline agent, demonstrating the strength of InternAgent-MLE. Furthermore, the paper compares the performance of InternAgent-MLE using different LLMs for **image, text, and tabular tasks**, proving its adaptability to different LLMs. Additionally, **ablation experiments** are conducted on the three main components of the new method, demonstrating their effectiveness and necessity. The appendix also provides a detailed supplement to the method’s functions, including **hyperparameter values for method calls and performance on various tasks**, mentioning the functional preferences of different LLMs and the agent’s ability to stimulate their strengths.

**Strengths:**

**Originality:**
This paper pioneers the integration of **Monte Carlo Graph Search (MCGS)** into the search strategy of **large language model (LLM) agents**, complemented by a **curated knowledge base** and **parallel exploration** mechanisms. Each key component is meticulously designed to address concrete challenges in Machine Learning Engineering (MLE), such as **node isolation** in traditional tree-based searches and **cold-start inefficiencies**. The solutions are well-justified with empirical evidence, yielding distinctive results (e.g., 36.4% average medal rate) and demonstrating high originality.
**Quality:**
In the experimental section, the paper evaluates multiple methods across various metrics on the authoritative **MLE-Bench** dataset. It conducts **ablation studies** on critical components (e.g., knowledge base, MCGS component) and tests the adaptability of **InternAgent-MLE** to different LLM backends (e.g., DeepSeek-R1, Gemini-2.5-pro). The methodology is rigorous, results are clear and reproducible, and the comparative analysis underscores the superiority of the proposed framework.
**Clarity:**
The paper provides a concise yet comprehensive explanation of **MCGS’s operational logic**, including its expansion mechanisms (e.g., intra-branch evolution, cross-branch reference) and edge types (primary vs. reference edges). Experimental hyperparameters (e.g., UCT constant, parallel workers) and implementation details are thoroughly documented (Table 4), ensuring high transparency and reproducibility.
**Significance:**
This work addresses the critical limitation of **branch isolation** in Monte Carlo Tree Search (MCTS) by enabling **trajectory reuse** and **cross-branch knowledge sharing**. It significantly boosts success rates on low-complexity tasks (62.1% vs. 48.5% for the best baseline) while reducing task completion time by **66%** (12 hours vs. 36 hours for Neo). The balance between **high-quality output** and **efficiency** makes InternAgent-MLE a powerful tool for MLE tasks. The detailed parameter configurations and benchmark results provide invaluable resources for future research.

**Weaknesses:**

The paper's description of **Monte Carlo Graph Search** is concise, which effectively communicates core concepts to domain experts. However, this brevity may pose challenges for **cross-disciplinary readers**. Including more detailed operational logic and illustrative diagrams of node interactions in the appendix could enhance accessibility for a broader audience.
Additionally, the results in **Table 1** reveal that **InternAgent-MLE** underperforms compared to **Neo** on **medium-complexity tasks**, despite demonstrating strong performance on both **low-complexity** and **high-complexity** tasks (Figure 4). This counterintuitive trend raises questions about the underlying causes. A discussion analyzing why performance specifically declines in **medium-difficulty tasks** would significantly improve the method's **theoretical coherence**.
There is an extraneous ")" at the end of line 195.

**Questions:**

The results in Table 1 show that InternAgent-MLE's performance falls slightly behind Neo on Medium Task Complexity tasks, yet it demonstrates superior performance on both Low and High Task Complexity tasks (as further supported by Figure 4). This observation raises a question: Intuitively, if a method's advancements enable it to outperform a baseline at both the lower and higher bounds of difficulty, it should logically maintain its lead on the intermediate difficulty level as well. The current result of underperformance on the medium level seems counter-intuitive. Could you please provide an explanation for this specific performance dip at the medium complexity level?

---

> ### Author Response · Authors · 2025-11-23
> **Official Comment to Reviewer bkGv**
>
> Thank you for taking the time to carefully review our paper. We appreciate your constructive feedback and valuable suggestions. Below, we provide detailed responses to each of your comments.
> > **[W1] Including more detailed operational logic and illustrative diagrams of node interactions in the appendix could enhance accessibility for a broader audience.**
>
> We appreciate the reviewer for the thoughtful suggestion and for highlighting that our description of Monte Carlo Graph Search (MCGS), while concise and accessible to domain experts, may be less friendly to cross-disciplinary readers.
>
> To improve clarity and reproducibility, we have made the following enhancements in the revised manuscript:
>
> - **Added a new section in** **Appendix A.2 and A.6** with a more complete explanation of the operational workflow, including illustrative diagrams depicting various types of expansion edges and example cases showing the agent’s decision logic during graph construction.
> - **Updated the main text** with additional clarifications on the operator selection mechanism and the specific conditions under which each expansion edge type is triggered in **Section 3.3**.
>
> We believe these additions substantially improve MCGS’s accessibility for readers without a background in search or planning and enhance the transparency and interpretability of our method. We sincerely appreciate the reviewer’s constructive feedback.
>
> > **[W2, Q1] The performance dip on medium-complexity tasks compared to Neo.**
>
> Thank you for raising this insightful and important point. We agree that the pattern observed in Table 1 and Figure 4 may initially appear counter-intuitive: InternAgent-MLE outperforms Neo on both low-complexity and high-complexity tasks, yet shows slightly weaker performance on the medium-complexity tier. We analyzed this phenomenon further and identified the following underlying factors:
>
> 1. **Medium-complexity tasks constitute the largest portion of the benchmark and place the strongest combined demand on both code-generation ability and domain understanding.** The limitations of the current base LLM are most evident in this middle region. In the medium-complexity category of MLE-Bench, many tasks require both a certain level of structured code generation (such as modular implementations or data-processing logic) and sufficient domain knowledge (such as image segmentation). These tasks differ from the Easy category, where the base model can often quickly infer a correct or nearly correct solution and obtain significant gains with the help of MCGS; they also differ from the Hard category, where the tasks exceed the reasoning capacity of the base model, allowing InternAgent-MLE to compensate more strongly for its weaknesses and achieve more noticeable improvements. However, medium-complexity tasks lie between these two extremes and rely more heavily on the base model’s native capabilities in “code reasoning” and “domain knowledge.” We observe that current LLMs indeed show certain gaps in this area, which results in smaller performance gains compared with the Easy and Hard ranges.
> 2. **Neo adopts a multi-model ensemble strategy that allows it to compensate for weaknesses of the underlying model on medium-complexity tasks.** By leveraging models such as GPT-5 and Claude 4.5, Neo benefits from stronger robustness in domain knowledge, problem interpretation, and Python code generation, which gives it an advantage specifically in this intermediate difficulty range. However, this comes with substantially higher cost and computational overhead.
>
> We will explore more fine-grained operator designs and stronger base-model adaptation mechanisms to reduce this gap in future iterations of InternAgent-MLE. We sincerely appreciate the reviewer’s careful observation, which helps strengthen the theoretical consistency and interpretation of our method.
>
> > **[W3] There is an extraneous ")" at the end of line 195.**
>
> We appreciate the reviewer for pointing out this formatting issue. After checking, we confirm that the “)” at the end of line 195 was indeed a typographical error, and it has been removed in the revised version. We have also carefully re-checked other paragraphs to ensure that all symbols and formatting are consistent. Thank you again for your careful reading.

---

### Author Response · Authors · 2025-12-01
**Author Comment to Area Chair**

Dear Area Chair,

We thank the Area Chair and all reviewers for their valuable time and feedback. In response to the reviewers’ concerns, we carried out extensive experiments and provided detailed clarifications. Although only Reviewer WoZe responded, he/she expressed agreement with our revisions and raised their score from 4 to 6. We hope that our rebuttal has also addressed the concerns of the other reviewers, and we will continue our efforts and look forward to contributing more broadly to the AutoMLE community.

To facilitate your assessment, we provide a concise overview of how we have addressed the key concerns raised across all reviewers. In our rebuttal, we offered detailed explanations of the core mechanisms of MCGS, the structure and usage of the knowledge base, and the overall evaluation setup. We additionally conducted a number of supplementary experiments to support these clarifications (**full results are included in the rebuttal**). Based on all reviewer feedback, we have made corresponding revisions throughout the manuscript.

### Reviewer bkGv:
- **On clarifying the operational logic and node interactions:**
We expanded **Appendix A.2 and A.6** with clearer workflow explanations and added illustrative diagrams depicting different reference edges and the planning process during graph construction. **Section 3.3** was refined to clarify the operator-selection mechanism.

- **On explaining the medium-complexity performance dip:**
We analyzed this issue and explained that Neo performs better in this range primarily because it relies on a multi-model ensemble of stronger LLMs (e.g., GPT-5, Claude 4.5), whose complementary strengths improve robustness on these tasks; however, this requires substantially higher cost and the use of more models.

### Reviewer tcRV:
- **On clarifying novelty and comparison with traditional MCGS:**
We added **Section 2.3** to provide a clear comparison with traditional MCGS and other graph-based search methods, explicitly highlighting how our formulation differs from prior work and clarifying the distinct strengths of MCGS in LLM-based coding scenarios (Response Part 1–2).

- **On refining operator logic and reward propagation:**
We provided a clearer and more concise explanation of the operator-selection mechanism on **pages 6–7** and revised **Section 3.3.2** with a more complete description of reward design and propagation, clarifying how operator choice depends on node state and search progress and how rewards are updated across parent nodes (Response Part 1).

- **On validating MCGS beyond MLE-Bench:**
We further demonstrated the generality of our MCGS framework by validating its effectiveness on additional coding tasks in the Autonomous Scientific Research domain (Response Part 2).
### Reviewer yhRj:
- **On strengthening comparisons and clarifying core mechanisms:**
We added **Section 2.3** to provide clearer comparisons with graph-based MCTS methods and highlighted how our method differs from prior approaches. We also refined **Page 6** with explicit descriptions of the triggering conditions for the three expansion-edge operators and added a clearer definition of the reward function on **Pages 6–7**, improving transparency of operator behavior and reward propagation (Response Part 1–3).

- **On illustrating Knowledge Base design:**
We provided detailed explanations of KB construction, decontamination procedures, and its internal organization in the rebuttal, along with a concrete illustrative example in **Appendix A.1**(Response Part 2).

- **On supplying ablations and sensitivity analyses:**
We included three-level internal KB ablations in **Appendix A.5** and added parameter-sensitivity experiments with corresponding analyses in **Appendix A.9** (Response Part 3).
### Reviewer WoZe:
- **On comparisons and MCGS mechanism clarity:**
We added extensive discussion and supplementary experiments comparing our method with MLE-STAR on MLE-Bench-Lite, and expanded the explanation of MCGS expansion and update behaviors, including concrete examples in **Appendix A.6** (Response Part 1–3).

- **On design of Knowledge Base:**
We supplied detailed clarification of KB size, construction process, and human annotation cost in the rebuttal, and added representative KB samples in **Appendix A.1** to improve transparency (Response Part 2).

- **On generalization beyond Kaggle-style tasks:**
We validated the robustness of our approach by conducting additional experiments in the Autonomous Scientific Research domain, demonstrating strong transferability beyond MLE-Bench (Response Part 3).

- **On computation cost and failure-case analysis:**
We provided further results and discussion on computational cost and failure cases in **Page 9** and **Page 25**, clarifying efficiency characteristics and system limitations (Response Part 2–4).

We trust that this summary clarifies how the reviewers’ concerns have been fully addressed. We sincerely appreciate your time and consideration.

---

### Meta-Review · Area_Chair_SHmq · 2026-01-05

**Summary:**

This paper presents **InternAgent-MLE**, an autonomous agent designed for Machine Learning Engineering (MLE) tasks. The core contribution is a **Monte Carlo Graph Search (MCGS) framework**, which extends Monte Carlo Tree Search by enabling cross-branch information sharing through reference edges, complemented by a curated ML knowledge base that provides task-agnostic priors. Reviewers emphasized the paper’s practical relevance and strong empirical system design, showing clear gains under a cost-efficient budget, and it appears robust across multiple task types. At the same time, they raised concerns that the technical novelty of MCGS is largely incremental over MCTS and lacks theoretical grounding, and that empirical comparisons, especially on medium-complexity tasks and against Neo, do not fully establish a clear algorithmic advantage.

**Reviewer Concerns:**

During the rebuttal phase, the authors addressed several reviewer concerns. They clarified the operational logic and triggering conditions of MCGS, discussed potential data contamination issues and the construction of the knowledge base, and provided additional evidence of cross-domain generalization, particularly in scientific research settings. These clarifications resolved some implementation-level ambiguities raised by the reviewers.

However, several concerns remain unresolved:
- **Limited Technical Novelty**: While the graph structure facilitates information reuse, the fundamental search paradigm remains an incremental extension of MCTS. (Reviewers tcRV, yhRj, WoZe).The method relies on heuristic triggers without a formal theoretical justification for why graph-based search improves convergence over MCTS. (Reviewer WoZe)
- **Insufficient empirical evidence and rigor (Reviewers bkGv)**: The empirical results do not demonstrate a clear algorithmic advantage over the Neo baseline in medium-complexity tasks and only reach parity in high-complexity scenarios. Furthermore, the use of different base models across baselines complicates fair comparison. The authors’ rebuttal attempts to attribute the medium-task performance gap to Neo’s use of stronger base models (e.g., GPT-5 and Claude 4.5), but this explanation is inconsistent with the experimental setup reported in Table 1, which uses GPT-4.1 and Claude 4. This inconsistency weakens the logical defense and raises concerns about whether the proposed framework can fundamentally outperform existing multi-agent architectures.

**Reviewer Scores:**

At the start of the rebuttal phase, two reviewers (tcRV and WoZe) recommended weak rejection (score 4), while the other two reviewers (bkGv and yhRj) expressed weak acceptance (score 6). Although the authors noted that one reviewer improved the score before the cutoff, the rebuttal does not sufficiently address the core concerns about novelty, theoretical contribution, and experimental rigor.

---

### Decision · Program_Chairs · 2026-01-26

Reject